# Generalizing Coverage Plots for Simulation-based Inference

**Maximilian Lipp**                                                                   *m.lipp@arcnl.nl*
*Science Park 106, 1098 XG Amsterdam, The Netherlands*
*Advanced Research Center for Nanolithography (ARCNL)*
*Vrije Universiteit Amsterdam*

**Benjamin Kurt Miller**                                                              *b.k.miller@uva.nl*
*Science Park 904, Amsterdam, The Netherlands*
*Informatics Institute, Institute of Physics*
*University of Amsterdam*

**Lyubov V. Amitonova**                                                              *l.amitonova@arcnl.nl*
*Science Park 106, 1098 XG Amsterdam, The Netherlands*
*Advanced Research Center for Nanolithography (ARCNL)*
*Department of Physics and Astronomy*
*Vrije Universiteit Amsterdam*

**Patrick Forré**                                                                     *p.d.forre@uva.nl*
*Science Park 904, Amsterdam, The Netherlands*
*Informatics Institute*
*University of Amsterdam*

**Reviewed on OpenReview:** *https://openreview.net/forum?id=eqYapbG2jO*

## Abstract

Simulation-based inference (SBI) aims to find the probabilistic inverse of a non-linear function by fitting the posterior with a generative model on samples. Applications demand accurate uncertainty quantification, which can be difficult to achieve and verify. Since the ground truth model is implicitly defined in SBI, we cannot compute likelihood values nor draw samples from the posterior. This renders two-sample testing against the posterior impossible for any practical use and calls for proxy verification methods such as expected coverage testing. We introduce a differentiable objective that encourages coverage in the generative model by parameterizing the dual form of the total variation norm with neural networks. However, we find that coverage tests can easily report a good fit when the approximant deviates significantly from the target distribution and give strong empirical evidence and theoretical arguments why the expected coverage plot is, in general, not a reliable indicator of posterior fit. To address this matter, we introduce a new ratio coverage plot as a better alternative to coverage, which is not susceptible to the same blind spots. It comes at the price of estimating a ratio between our model and the ground truth posterior, which can be done using standard algorithms. We provide experimental results that back up this claim, and provide multiple algorithms for estimating ratio coverage.

## 1 Introduction

We are concerned with a perennial question in Simulation-Based Inference (SBI): *How do we determine if our learned approximation to the posterior accurately represents the ground truth?*

Recall that the necessary ingredients for deep learning-based SBI include: a non-linear, stochastic function, known as a simulator, that takes in parameters $\theta$ and returns a simulated observation $x$; a prior distribution over all possible parameters $p(\theta)$; and a method that infers the distribution over parameters that would plausibly generate $x_o$ if they were passed to the simulator. This target distribution is called the posterior $p(\theta \mid x) := \frac{p(x|\theta)}{p(x)} p(\theta)$ evaluated at observation $x_o$, namely $p(\theta \mid x_o)$. In SBI, we typically cannot evaluate the likelihood $p(x \mid \theta)$, so we need to estimate the posterior using a generative model.

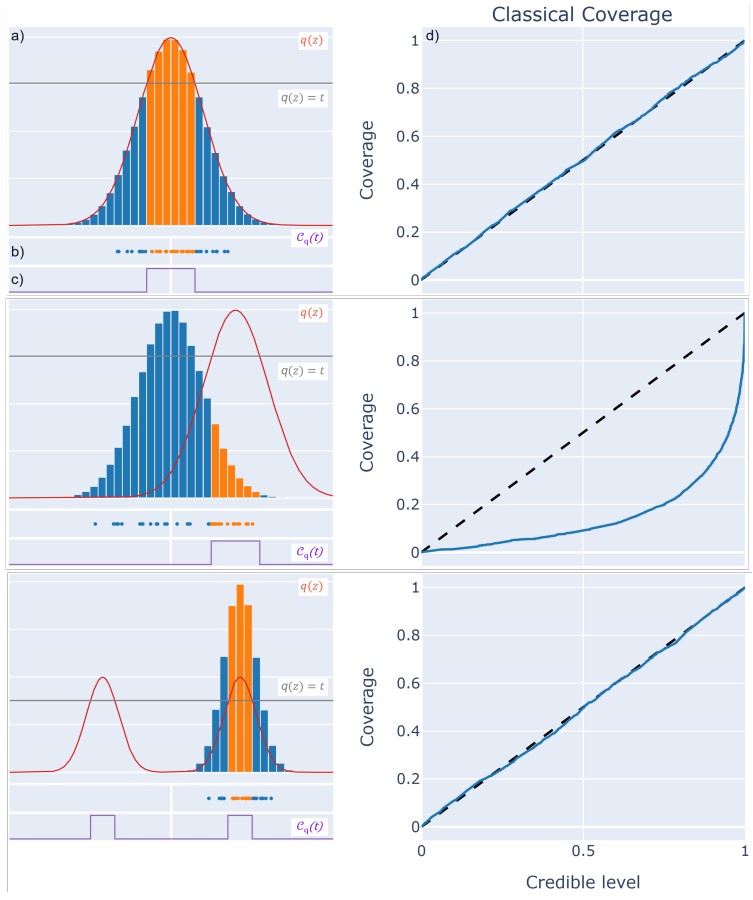

Figure 1: A one-dimensional example highlighting the problem of *classical coverage*, which cannot separate the very different model and true distribution. The *classical coverage* is calculated using 1D normal distributed data $p(z)$ and different models $q(z)$. Subplot a) shows the histogram of the data as well as the learned model distribution $q(z)$. The true data distribution $p(z)$ is not shown since it is not available in practice. In subplot b), some of the original 1D data points $z \sim p(z)$ are shown, where orange points are selected by the threshold $q(z) \geq t$ and blue data are not. To calculate the *classical coverage*, first a threshold $t$ is chosen on the model distribution $q(z)$, which leads to the super-level set $\mathcal{C}_q(t)$ illustrated in subplot c). Then for subfigure d), the distribution $p(z)$ is integrated in the region of the super-level set $P(\mathcal{C}_q(t))$ (using the histogram data) and plotted against the integrated distribution $Q(\mathcal{C}_q(t))$ for many different super-level sets with corresponding thresholds $t$. Since the model $q(z)$ closely matches the histogram $p(z)$, the coverage correctly shows a diagonal line. In the top row, the model and data distribution agree. In the middle row, the model is shifted to the side and in the bottom row the model has two peaks, which also creates perfect coverage.

The inference method produces an approximation to the posterior $q(\theta \mid x)$. Since the true posterior $p(\theta \mid x)$ is unknown for any practical use case, it is not straightforward to determine whether our approximation accurately represents the ground truth using two-sample testing (Gretton et al. (2006); Lopez-Paz & Oquab (2017); Friedman (2004)). An ideal algorithm would compare $q(\theta \mid x_o)$ to $p(\theta \mid x_o)$ without evaluating or sampling the posterior. Since such a comparison is impossible in practice, recent papers instead demonstrate

a proposed inference method using two-sample tests on toy problems with known solutions Lueckmann et al. (2021). This does not help practitioners who need to know whether their specific posterior approximation can be trusted, not the performance of the inference algorithm on a toy problem.

In lieu of this ideal, fictional algorithm, practitioners typically test the *expected conditional coverage* of their approximation (Miller et al. (2021); Hermans et al. (2022); Talts et al. (2020); Zhao et al. (2021)). Intuitively, expected conditional coverage tests whether the approximation's credible region cover the ground truth's credible region, at all credible levels, averaged over $x \sim p(x)$. Although this sounds promising as a criterion to determine whether two distributions agree, there are issues with expected conditional coverage. One known limitation is that the prior $p(\theta)$ has exact expected conditional coverage at all credibility levels Delaunoy et al. (2023); Lemos et al. (2023), i.e. this method cannot distinguish between $p(\theta)$ and $p(\theta \mid x)$. Despite this limitation, many works have proposed algorithms aimed at improving the expected conditional coverage of learned posteriors through regularization (Delaunoy et al. (2022; 2023); Dey et al. (2022); Falkiewicz et al. (2023)). We will propose another one.

We find an additional, significant blind spot in expected conditional coverage that has gone hithertofore unmentioned in literature: *Coverage does not always penalize the approximant for putting mass in regions where the ground truth has none!* Note that this limitation is quite deep in the notion of coverage itself, extending to the so-called *unconditional coverage*, i.e. coverage testing between arbitrary distributions $p(z)$ and $q(z)$ that is not averaged over a conditional variable. We visualize an example of such a failure mode in Figure 1. In addition to this pathological example, we create a differentiable objective that encourages unconditional coverage on $q(\theta \mid x)$ for every $x$, and thereby expected conditional coverage, and find that we can produce distributions with good expected conditional coverage, even though the approximation and the posterior significantly disagree.

To address this issue, we propose a generalized version of coverage that (a) discriminates between distributions $q$ and $p$, i.e. it only returns the typical diagonal line when $q = p$, (b) does not require samples from the ground truth posterior, and (c) does not suffer from either blind spot mentioned above. However, it comes at the cost of estimating the ratio $\frac{q(\theta|x)p(x)}{p(\theta,x)} = \frac{q(\theta|x)}{p(\theta|x)}$.

**Contribution** (1) We formulate a rigorous definition for both unconditional and expected conditional coverage. We relate coverage tests to divergences between probability measures and show that the typical learning objective for Neural Posterior Estimation (NPE) penalizes a lack of coverage. (2) Propose a differentiable and adversarial regularization objective based on the total variation distance that encourages coverage and show that unconditional coverage is not discriminative, i.e., when $q$ has unconditional coverage w.r.t. $p$ that *does not* imply $p = q$. We thereby address a commonly held misconception that issues with expected conditional coverage only stem from the necessity to average over $p(x)$. (3) Present an alternative form of coverage called the *ratio coverage* which *is* discriminative. We formulate it both unconditionally, over the joint distribution of $\theta$ and $x$, and as expected conditional ratio coverage. (4) Provide experimental evidence for (2) and (3).

**Related Work** We addressed the relevant two-sample tests and coverage testing for SBI above; however, we specifically mention the work $\ell$-C2ST Linhart et al. (2023), for sequential processes SSNL Dirmeier et al. (2024). $\ell$-C2ST, which is more precise and computationally efficient than its concurrent method local-HPD Zhao et al. (2021), is similar to ratio coverage because we also estimate a ratio between $q(\theta \mid x)$ and $p(\theta \mid x)$ using the likelihood ratio trick (Hermans et al. (2020); Durkan et al. (2020); Miller et al. (2022); Gutmann & Hyvärinen (2010); Oord et al. (2018); Dalmasso et al. (2020)) or other methods (Miller et al. (2023); Federici et al. (2023); Nguyen et al. (2010); Yao & Domke (2023)). $\ell$-C2ST and the HPD methodology summarizes the whole information concerning $\theta$ into a single scalar or a $\theta$-vector respectively, while this work uses an estimator to improve the statistical power of the well established coverage plot. So instead of proposing a new method for inspecting local posterior consistency, we show that the widely used coverage plot can be modified to achieve full discriminative power. An estimator with ratio coverage would pass $\ell$-C2ST for all $x_o$.

## 2 Preliminaries

Since this paper deals with the fundamental properties of coverage as a measurement of similarity between probability distributions, we treat the problem from a fundamental perspective. In particular, we will treat the properties of unconditional coverage using on a general probability space $\mathcal{Z}$, which we take to be the joint space of parameters and data $\Theta \times \mathcal{X}$ in the SBI case. Furthermore, critical to many of our practical results for SBI is the asymmetry between ground truth posterior $p$ and approximation $q$. We typically assume we can neither evaluate nor sample from $p(\theta \mid x)$, but we can do both with $q(\theta \mid x)$. This is the normal situation for expected coverage testing.

Throughout, we deal with comparing distributions and we seek to formalize this using the following divergences and measures.

**Definition 2.1** (Kullback-Leibler divergence)**.** *Let $(\mathcal{Z}, \mathcal{B}_{\mathcal{Z}})$ be a measurable space and $P$ and $Q$ two probability measures on $(\mathcal{Z}, \mathcal{B}_{\mathcal{Z}})$ with densities $p$ and $q$. The* Kullback-Leibler divergence *is then:*

$$\mathrm{KL}(P\|Q) := \mathbb{E}_P\left[\log\left(\frac{p}{q}\right)\right]. \tag{1}$$

The Kullback-Leibler divergence is a common statistical distance, often used as an NPE training objective and also forms the foundation for our regularizer (see Section 5).

**Definition 2.2** (Total variation distance/norm)**.** *Let $(\mathcal{Z}, \mathcal{B}_{\mathcal{Z}})$ be a measurable space and $P$ and $Q$ two probability measures on $(\mathcal{Z}, \mathcal{B}_{\mathcal{Z}})$. We then define their* total variation distance/norm *by the formula:*

$$\mathrm{TV}(P\|Q) := \sup_{A \in \mathcal{B}_{\mathcal{Z}}} |P(A) - Q(A)|, \tag{2}$$

*which is always a number in the interval $[0, 1]$.*

**Theorem 2.3** (For a proof see Theorem A.6)**.** *We have the following identities for the total variation norm:*

$$\mathrm{TV}(P\|Q) = \frac{1}{2}\int_{\mathcal{Z}} |p(z) - q(z)| \, \nu(dz) = \frac{1}{2} \sup_{\substack{f:\mathcal{Z}\to[-1,1] \\ measurable}} |\mathbb{E}_P[f] - \mathbb{E}_Q[f]|, \tag{3}$$

*where $p$ and $q$ are densities for $P$ and $Q$, resp., w.r.t. any fixed joint dominating measure $\nu$, e.g. $\nu = \frac{1}{2}(P+Q)$. Furthermore, the supremum on the rhs is attained for the map $f^\star$ given as follows:*

$$f^\star(z) := \mathrm{sgn}\log\left(\frac{q(z)}{p(z)}\right) = \begin{cases} 1 & \text{if } q(z) > p(z), \\ 0 & \text{if } q(z) = p(z), \\ -1 & \text{if } q(z) < p(z). \end{cases} \tag{4}$$

The Total variation distance is also a statistical distance between probability distributions, which we can relate to an optimality condition on the *classical coverage* and subsequently results in the *adversarial total variation norm regularizer* (see Section 5).

## 3 Generalized Coverage Plots

We define the fundamental terms analogous to the choice from Hermans et al. (2022), but split the derivation into two steps: first introducing the mathematical simpler case of *unconditional coverage* and then extending to the more widely used definition of *expected conditional coverage*. This generalized framework allows us to investigate different kinds of coverage plots, e.g. the *classical coverage plot* in Example 3.3.

### 3.1 Generalizing Unconditional Coverage Plots

We start with a generalized definition of *coverage* for unconditional probabilities, then explain the specific choice leading to the *classical coverage* and finally focuses on estimating these coverage plots.

**Definition 3.1** (Unconditional coverage plots and coverage gap)**.** *Let $(\mathcal{Z}, \mathcal{B}_{\mathcal{Z}})$ be a measurable space, $P$ and $Q$ two probability measures on that space, and, let $g : \mathcal{Z} \to \bar{\mathbb{R}} := \mathbb{R} \,\dot{\cup}\, \{\pm\infty\}$ be a measurable map. The* (unconditional) coverage plot *of $P$ and $Q$ w.r.t.* discriminating function $g$ is defined to be (an estimate) of *the following graph:*

$$\Gamma_g(P,Q) := \left\{ (Q(\mathcal{C}_g(t)), P(\mathcal{C}_g(t))) \,\middle|\, t \in \bar{\mathbb{R}} \right\} \subseteq [0,1] \times [0,1], \tag{5}$$

*where $\mathcal{C}_g(t)$ is the super-level set of $g$ for threshold $t \in \bar{\mathbb{R}}$:*

$$\mathcal{C}_g(t) := \{z \in \mathcal{Z} \,|\, g(z) \geq t\} = g^{-1}([t, \infty]). \tag{6}$$

*Furthermore, we define the* (unconditional) coverage gap *between $P$ and $Q$ w.r.t. $g$ as follows:*

$$\Delta_g(P\|Q) := \sup_{t \in \bar{\mathbb{R}}} |P(\mathcal{C}_g(t)) - Q(\mathcal{C}_g(t))| \in [0,1]. \tag{7}$$

*For further information, see Proposition A.4.*

**Remark 3.2** (Estimating the coverage plot with samples)**.** *Assume that we have an i.i.d. sample $\{z_1, \ldots, z_N\}$ from $P$ and an i.i.d. sample $\{z_1', \ldots, z_M'\}$ from $Q$. Then we can approximate the coverage plot $\Gamma_g(P,Q)$ as follows. First consider the set of all relevant thresholds:*

$$\mathcal{T} := \{g(z_n) \,|\, n \in [N]\} \,\dot{\cup}\, \{g(z_m') \,|\, m \in [M]\}. \tag{8}$$

*Then for every $t \in \mathcal{T}$ we compute:*

$$\hat{p}(t) := \# \{n \in [N] \,|\, g(z_n) \geq t\} \,/N, \qquad \hat{q}(t) := \# \{m \in [M] \,|\, g(z_m') \geq t\} \,/M, \tag{9}$$

*and plot the corresponding points for all $t \in \mathcal{T}$:*

$$\hat{\Gamma}_g(P,Q) := \{(\hat{q}(t), \hat{p}(t)) \,|\, t \in \mathcal{T}\}. \tag{10}$$

*See Algorithm 1 and Figure 1 for more information.*

The coverage gap $\Delta_g(P\|Q)$, more intuitively, measures the maximal distance between the coverage plot $\Gamma_g(P,Q)$ and the diagonal $\Delta_{[0,1]} = \{(r,r) \,|\, r \in [0,1]\}$ of the square $[0,1] \times [0,1]$.

**Example 3.3** (Classical coverage plot)**.** *We regain the* classical coverage plot *for the function $g(z) := q(z)$, the probability density of the model distribution $Q$. Then, the super-level sets correspond to the highest probability confidence regions of $q(z)$. This choice for $g(z)$ comes however with a severe blind spot, because it cannot generally distinguish between $P$ and $Q$, i.e., it does not detect any probability mass in a region where $q(z)$ has small probability. Figure 1 shows that the* classical coverage gap *vanishes: $\Delta_q(P\|Q) = 0$, although $P \neq Q$. A more rigorous proof can be found in Example A.9.*

---

**Algorithm 1** Generalized unconditional coverage plot

---

**Require:** simulator $p(x|\theta)$, prior $p(\theta)$, surrogate model $q(\theta|x)$, discriminating function $g(\theta, x)$, sample size numbers $N$, $M$;
**Ensure:** list of confidence level pairs $\{(\gamma_n, \gamma'_n) \,|\, n \in [N]\}$;
  **for** $m = 1$ to $M$ **do**
    $\theta_m \sim p(\theta)$
    $x_m \sim p(x|\theta_m)$
    $\hat{\theta}_m \sim q(\theta|x_m)$
  **end for**
  **for** $n = 1$ to $N$ **do**
    $\theta_n \sim p(\theta)$
    $x_n \sim p(x|\theta_n)$
    $\gamma_n \leftarrow \frac{1}{M} \sum_{m=1}^{M} \mathbb{1}[g(\hat{\theta}_m, x_m) \geq g(\theta_n, x_n)]$
  **end for**
  **for all** $\gamma_k$ in **sort**$(\{\gamma_n \,|\, n \in [N]\})$ **do**
    $(\gamma_k, \gamma'_k) \leftarrow (\gamma_k, \frac{k}{N})$
  **end for**

---

### 3.2 Generalizing Expected Conditional Coverage Plots

Next, we follow again three steps for the *expected conditional coverage*: giving a generalized definition, discuss the *expected conditional classical coverage* widely used in literature and finally talk about practical implications for optimization.

**Definition 3.4** (Expected conditional coverage plots and coverage gap). *Let $(\Theta, \mathcal{B}_\Theta)$ and $(\mathcal{X}, \mathcal{B}_\mathcal{X})$ be measurable spaces, let $P = P(\Theta, X)$ and $Q = Q(\Theta, X)$ be two probability measures on $\mathcal{Z} := \Theta \times \mathcal{X}$ with the same marginal $P(X) = Q(X)$[1], and let $g : \Theta \times \mathcal{X} \to \bar{\mathbb{R}} := \mathbb{R} \,\dot{\cup}\, \{\pm\infty\}$ be a measurable map. First, consider the* conditional survival function $S$ *of $g$ w.r.t $Q$, which is on $t \in \bar{\mathbb{R}}$, $x \in \mathcal{X}$ and $\theta \in \Theta$ given by:*

$$S(t|x) := Q(g(\theta, x) \geq t | X = x), \tag{11}$$

*Its partial inverse, the* threshold function *is given for $\gamma \in [0, 1]$ and $x \in \mathcal{X}$ as follows:*

$$t_\gamma(x) := \sup\left\{ t \in \bar{\mathbb{R}} \,\big|\, S(t|x) \geq \gamma \right\} \in \bar{\mathbb{R}}. \tag{12}$$

*With this we define the set:*

$$\mathcal{C}_g(\gamma) := \{(\theta, x) \in \Theta \times \mathcal{X} \,|\, g(\theta, x) \geq t_\gamma(x)\}. \tag{13}$$

*The* expected conditional coverage plot *of $P$ and $Q$ w.r.t. $g$ is defined to be (an estimate) of the following graph:*

$$\bar{\Gamma}_g(P, Q) := \{(Q(\mathcal{C}_g(\gamma)), P(\mathcal{C}_g(\gamma))) \,|\, \gamma \in [0, 1]\} \subseteq [0, 1] \times [0, 1]. \tag{14}$$

*Furthermore, we define the* expected conditional coverage gap *between $P$ and $Q$ w.r.t. $g$ as follows:*

$$\bar{\Delta}_g(P\|Q) := \sup_{\gamma \in [0,1]} |P(\mathcal{C}_g(\gamma)) - Q(\mathcal{C}_g(\gamma))| \in [0, 1]. \tag{15}$$

Before estimating the *expected conditional coverage plot*, note that for every $x \in \mathcal{X}$ and $\gamma \in [0, 1]$ we always have:

$$Q(g(\Theta, x) \geq t_\gamma(x)|X = x) \geq \gamma, \qquad\qquad Q(\mathcal{C}_g(\gamma)) \geq \gamma, \tag{16}$$

---

[1]In SBI, the simulated data $P(X)$ and the input data to the model $Q(X)$ are both generated in the same way: $\theta \sim P(\Theta)$ and then $x \sim P(X|\Theta = \theta)$, hence $P(X) = Q(X)$.

with equalities under suitable continuity/positivity conditions on the conditionals $Q(\Theta|X = x)$. In those cases we are thus just plotting $P(\mathcal{C}_g(\gamma))$ against $\gamma \in [0, 1]$.

Additionally note that we have the following equivalence for $t \in \bar{\mathbb{R}}$ and $\gamma \in [0, 1]$ under suitable continuity/positivity conditions:

$$S(t|x) \leq \gamma \qquad \Longleftrightarrow \qquad t \geq t_\gamma(x). \tag{17}$$

**Remark 3.5** (Estimating the expected conditional coverage plot with samples). *For $n \in [N]$ sample $\theta_n \sim P(\Theta)$ and $x_n \sim P(X|\Theta = \theta_n)$ and put $t_n := g(\theta_n, x_n)$. For $m \in [M]$ further sample $\theta_{n,m} \sim Q(\Theta|X = x_n)$. Then put:*

$$\gamma_n := \frac{1}{M} \sum_{m=1}^{M} \mathbb{1}[g(\theta_{n,m}, x_n) \geq t_n] \approx Q(g(\Theta, x_n) \geq t_n|X = x_n) = S(t_n|x_n). \tag{18}$$

*Further define for $\gamma \in [0, 1]$:*

$$\hat{F}(\gamma) := \frac{1}{N} \sum_{n=1}^{N} \mathbb{1}[\gamma_n \leq \gamma] \overset{(18)}{\approx} \frac{1}{N} \sum_{n=1}^{N} \mathbb{1}[S(t_n|x_n) \leq \gamma] \tag{19}$$

$$\overset{(17)}{\approx} \frac{1}{N} \sum_{n=1}^{N} \mathbb{1}[t_n \geq t_\gamma(x_n)] \tag{20}$$

$$= \frac{1}{N} \sum_{n=1}^{N} \mathbb{1}[g(\theta_n, x_n) \geq t_\gamma(x_n)] \tag{21}$$

$$\approx P(g(\Theta, X) \geq t_\gamma(X)). \tag{22}$$

*With these approximations and the argument in (16) we can thus estimate the expected conditional coverage plot as:*

$$\hat{\Gamma}_g(P\|Q) := \left\{ \left(\gamma, \hat{F}(\gamma)\right) \,\Big|\, \gamma \in [0, 1] \right\} \subseteq [0, 1] \times [0, 1]. \tag{23}$$

*See Algorithm 2 and Figure 1 for more information.*

**Example 3.6** (Expected conditional classical coverage plot). *Analogue to Example 3.3, we regain the* classical coverage *for the choice $g(\theta, x) = \log q(\theta|x)$. In contrast to the previous Example, this plot first determines the coverage conditioned on $x$ over $\theta$, before then averaging the result over all $x$. Hence, the* expected conditional coverage *plot compares $p(\theta|x)$ to $q(\theta|x)$ while the* unconditional coverage *plot compares $p(\theta|x)p(x)$ to $q(\theta|x)p(x)$.*

**Remark 3.7.**

1. *We always have the trivial inequality:*

$$\bar{\Delta}_g(P\|Q) \leq \sup_{C \in \mathcal{B}_{\Theta \times \mathcal{X}}} |P(C) - Q(C)| = \mathrm{TV}(P\|Q), \tag{24}$$

   *where the supremum ranges over all measurable subsets $C$ of the product space $\Theta \times \mathcal{X}$.*

2. *Because we assumed that the marginals agree: $P(X) = Q(X)$, we also have the following tighter inequalities (full proof see Remark A.2):*

$$\bar{\Delta}_g(P\|Q) \leq \mathbb{E}_{P(X)} \left[ \mathrm{KS}(g_{\#}^X P(\Theta|X) \| g_{\#}^X Q(\Theta|X)) \right] \tag{25}$$

$$\leq \mathbb{E}_{P(X)} \left[ \mathrm{TV}(g_{\#}^X P(\Theta|X) \| g_{\#}^X Q(\Theta|X)) \right] \tag{26}$$

$$= \frac{1}{2} \sup_{\substack{f: \bar{\mathbb{R}} \times \mathcal{X} \to [-1, 1] \\ measurable}} |\mathbb{E}_P[f] - \mathbb{E}_Q[f]|, \tag{27}$$

with the push-forward probability measures $g_\#^X Q(\Theta|X)$ and $g_\#^X P(\Theta|X)$ *(Tao (2021)), KS the Kolmogorov–Smirnov divergence (Definition A.1) and $f = f(g(\Theta, X), X)$ where for $x \in \mathcal{X}$ we abbreviate the partially evaluated map:*

$$g^x: \Theta \to \bar{\mathbb{R}}, \qquad\qquad g^x(\theta) := g(\theta, x). \qquad (28)$$

---

**Algorithm 2** Generalized expected conditional coverage plot

---

**Require:** simulator $p(x|\theta)$, prior $p(\theta)$, surrogate model $q(\theta|x)$, discriminating function $g(\theta, x)$, sample size numbers $N$, $M$;

**Ensure:** list of confidence level pairs $\{(\gamma_n, \gamma_n') \,|\, n \in [N]\}$;

    **for** $n = 1$ to $N$ **do**
        $\theta_n \sim p(\theta)$
        $x_n \sim p(x|\theta_n)$
        **for** $m = 1$ to $M$ **do**
            $\theta_{n,m} \sim q(\theta|x_n)$
        **end for**
        $\gamma_n \leftarrow \frac{1}{M} \sum_{m=1}^{M} \mathbb{1}[g(\theta_{n,m}, x_n) \geq g(\theta_n, x_n)]$
    **end for**
    **for all** $\gamma_k$ in **sort**$(\{\gamma_n \,|\, n \in [N]\})$ **do**
        $(\gamma_k, \gamma_k') \leftarrow (\gamma_k, \frac{k}{N})$
    **end for**

---

## 4  The Ratio Coverage

As we pointed out that *classical coverage* tests cannot in general discriminate different distributions, we propose the *ratio coverage*, which does not have this deficiency, but still shares valuable properties with the classical coverage, e.g. does not require samples from the ground truth posterior. The *ratio coverage* is motivated by the supremum found in Theorem 2.3 which is $f^\star(z) = \operatorname{sgn} g(z)$ with the choice $g(\theta, x) := \log q(z)/p(z)$ (see Remark A.3 and Figure 2).

**Theorem 4.1** (The ratio coverage plots are discriminating)**.** *Let $P = P(\Theta, X)$ and $Q = Q(\Theta, X)$ be two probability measures on $\mathcal{Z} = \Theta \times \mathcal{X}$ with the same marginal $Q(X) = P(X)$ and with densities $p$ and $q$. Consider the (log) ratio function $g$ given as follows:*

$$g(\theta, x) := \log \frac{q(\theta, x)}{p(\theta, x)} = \log \frac{q(\theta|x)}{p(\theta|x)}. \qquad (29)$$

*Then both, the* unconditional coverage plot $\Gamma_g(P\|Q)$ *and the* expected conditional coverage plot $\bar{\Gamma}_g(P\|Q)$ *each can discriminate between $P$ and $Q$:*

$$\Delta_g(P\|Q) = 0 \implies P = Q, \qquad\qquad \bar{\Delta}_g(P\|Q) = 0 \implies P = Q. \qquad (30)$$

*Furthermore, we have the following (in)equalities for the* coverage gaps *and* TV *norms:*

$$\Delta_g(P\|Q) \leq \mathrm{TV}(g_\# P \| g_\# Q) = \mathrm{TV}(P\|Q); \qquad (31)$$

$$\bar{\Delta}_g(P\|Q) \leq \mathbb{E}_{P(X)}\left[\mathrm{TV}(g_\#^X P(\Theta|X) \| g_\#^X Q(\Theta|X))\right] = \mathrm{TV}(P\|Q). \qquad (32)$$

*Proof.* See Theorem A.7, Corollary A.8 and Figure 2. □

However, in a practical setting the *ratio coverage* cannot be calculated directly, because it requires the model distribution $q(z)$ as well as the true posterior distribution $p(z)$. Therefore, we propose to train a classifier to approximate the ratio $q(z)/p(z)$. This approach must be distinguished from Neural Ratio Estimation (NRE), which instead learns the likelihood-to-evidence ratio. Additionally, a single value metrics can also be

obtained by calculating the *total variation norm* between both distributions directly using the learned ratio as similarly proposed in Gutmann et al. (2018).

This ratio training is also a limitation of the ratio coverage, because of potentially large parameter space spanned by $\Theta \times \mathcal{X}$. The training requires additional computational cost, and the ratio coverage directly depends on an accurate estimation. However, the additional effort is justified by the full discriminating power, specifically in comparison to the classical coverage, which also requires intensive computing resources.

In summary, we proofed that the *ratio coverage* preserves full discriminating power for both plots, the *unconditional coverage plot* and the *expected conditional coverage plot*. The *unconditional coverage plot* calculates the coverage symmetrically over the joined space $\Theta \times \mathcal{X}$. In contrast, the *expected conditional coverage plot* first determines the coverage conditioned on $x$ over $\theta$, before then averaging the result over all $x$. Hence, the *expected conditional coverage plot* compares $p(\theta \mid x)$ to $q(\theta \mid x)$ while the *unconditional coverage plot* compares $p(\theta \mid x)p(x)$ to $q(\theta \mid x)p(x)$. The *unconditional coverage plot* is much more simpler in regards of derivation as well as in terms of computational cost, since it requires far fewer samples from the model. Although the expected conditional coverage has been mainly used throughout literature (Hermans et al. (2022)), the unconditional coverage has clear advantages in terms of simplicity and computing cost.

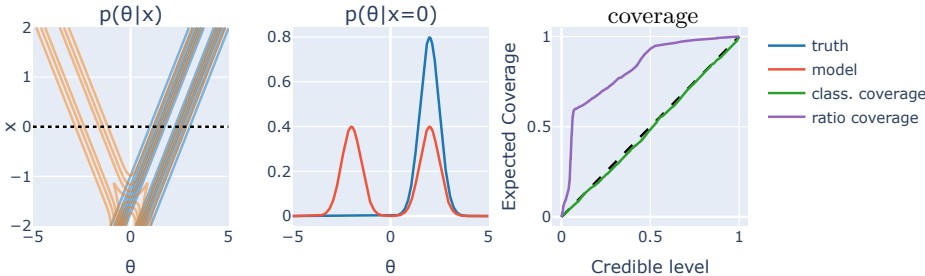

Figure 2: An example illustrating the problem of classical coverage for conditional probabilities. The true and model dist. given by $p(\theta|x)$ and $q(\theta|x)$ are shown in the first image and the cross section for $x = 0$ is presented in the second plot (black dotted line). Similar to the unconditional case, these distributions exhibit one and two Gaussian peaks, whose mean value is shifted by $x$. The third plot show the approximated classical and ratio coverage based on the Algorithm 2 for a finite data sets drawn from the distributions.

Looking at *ratio coverage* from the point of view of hypothesis testing, we consider a sample $z$ drawn from $P$ or $Q$, and our test should determine from which distribution it most likely originates. For some (summary) test statistic $g$ and threshold $t$, we decide for $P$, if $g(z) < t$ and decide for $Q$ if $g(z) > t$. In other words, the null hypothesis corresponds to $H_0 : P$ while the alternative hypotheses is $H_A : Q$. Our unconditional ratio coverage plot directly plots the following test quantities against each other: $P(g > t) = typeIerror$ and $Q(g > t) = 1 - typeIIerror =: power$. So our coverage plot has direct statistical interpretation and by the classical Neyman-Pearson lemma the unique uniformly most powerful test is the ratio test. So, with our approach, we learn the most powerful summary statistic.

## 5    Classical Coverage Regularization Can Be Deceptive

Because the *classical coverage* is widely viewed as a key performance metric, usually a next step would be to develop a regularizer to emphasize well calibrated models. In this section we focus on finding such a regularization objective to exemplify how this goes wrong because of the blind spot of the *classical coverage*. As shown in Figure 1, the *classical coverage* can be optimal, although the model did not learn the true distribution. We develop the Adversarial Total Variation (TV) Regularization as a differentiable upper bound to the *coverage gap*, which will lead to improved *classical coverage* in Section 6, but simultaneously create great disagreement between the learned and the true posterior distributions. This problem is then exacerbated by the fact that the *classical coverage* is unable to detect this specific kind of misalignment,

because of its blind spot. The same phenomenon can also be observed when regularizing with techniques from the literature (Section 6.3).

By using that the *expected conditional coverage gap* for general choice of $g$ is bounded by the *total variation norm*, we derive an adversarial regularizer for minimizing the coverage gap.

Equation (27) provides us with an adversary regularization bound for minimizing the *coverage gap*:

$$\bar{\Delta}_g(P\|Q) \leq \frac{1}{2} \sup_{\substack{f:\bar{\mathbb{R}}\times\mathcal{X}\to[-1,1] \\ \text{measurable}}} |\mathbb{E}_P[f] - \mathbb{E}_Q[f]| \tag{33}$$

with $f = f(g(\Theta, X), X)$.

Under certain regularity assumptions on the densities $p$ and $q$ we can restrict the model class $\mathcal{F}$ for the discriminator $f : \bar{\mathbb{R}} \times \mathcal{X} \to [-1, 1]$ to sufficiently flexible neural network classes with tanh-outputs such that the corresponding Universal Approximation Theorem holds, see Kurt Hornik (1991). We thus arrive at the regularizing adversarial objective for such neural network classes $f \in \mathcal{F}$:

$$\mathcal{R}_g(Q, f) = |\mathbb{E}_P[f \circ g] - \mathbb{E}_Q[f \circ g]|. \tag{34}$$

We can now combine this regularizer with the usual NPE training objective, the *Kullback-Leibler divergence* (Papamakarios et al. (2017)), resulting in the adversarial loss function with Lagrange multiplier $\lambda \geq 0$:

$$\mathcal{L}_g(Q, f) = \mathrm{KL}(P\|Q) + \lambda \cdot \mathcal{R}_g(Q, f), \tag{35}$$

where we maximize w.r.t. the parameters of $f$ and minimize w.r.t. the parameters of $Q$.

For the *expected conditional classical coverage*, i.e. $\mathcal{Z} = \Theta \times \mathcal{X}$ and $g(\theta, x) := \log q(\theta|x)$, we get the adversarial loss (written with densities and arguments):

$$\mathcal{L}(q, f) = -\mathbb{E}_{p(\theta,x)}[\log q(\theta|x)] + \lambda \cdot |\mathbb{E}_{p(\theta,x)}[f(\log q(\theta|x), x)] - \mathbb{E}_{q(\theta,x)}[f(\log q(\theta|x), x)]|, \tag{36}$$

where we, again, maximize w.r.t. the parameters of $f$ and minimize w.r.t. the parameters of $q(\theta|x)$.

Analogously to the adversary TV regularizer for the *classical coverage*, we could also investigate an adversarial ratio regularizer. But for the NPE objective Equation (35) it is important to note that for the log ratio $g = \log \frac{q}{p}$ we have the chain of inequalities (Pinsker's inequality) for the coverage gap:

$$\Delta_g(P\|Q)^2 \leq \mathrm{TV}(g_\# P\|g_\# Q)^2 = \mathrm{TV}(P\|Q)^2 \leq \frac{1}{2}\mathrm{KL}(P\|Q). \tag{37}$$

So, the usual NPE objective (the KL divergence) already closely upper bounds the adversarial ratio regularizer (Remark A.5), which can thus be dropped from the NPE training objective in this case. Similar arguments hold for the expected conditional version.

## 6 Experiments

In this section we present the empirical evidence for the problem with the TV regularization, the superior discrimination power of the *ratio coverage* and the comparison between *expected conditional* and *unconditional coverage plots*. In our experiments, we follow the procedure from Hermans et al. (2022) for evaluating SBI models on the SLCP. Specifics about the implementation and computational setup can be found in Section A.2 and additional results for the two moons benchmark task can be found in Section A.3.

Code is available: https://anonymous.4open.science/r/ratio_coverage-E777/.

### 6.1 Results

The first aspect of the evaluation of the models is the *expected conditional classical coverage* in Figure 3 for simulation budgets $2^{10}$ and $2^{16}$ with and without TV regularization. There is a clear correlation between

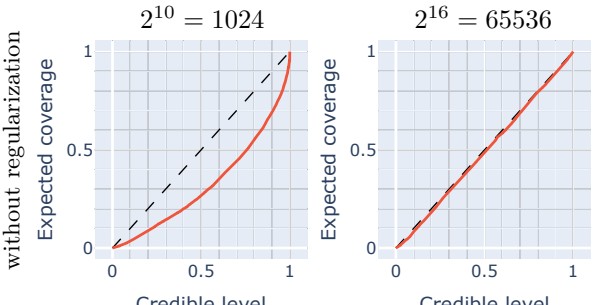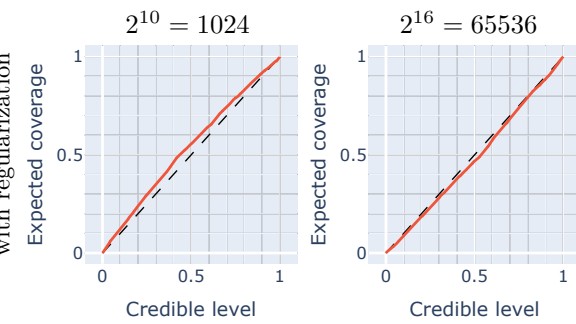

Figure 3: These plots indicate improvements due to the TV regularization for the simulation budgets $2^{10}, 2^{16}$ but this originates from the lack of discriminating power of the *expected conditional classical coverage*. The coverage plots are determined using Algorithm 2 ($M = N = 1024$).

better coverage and larger simulation budget. In the regularized case, this trend is not as clearly visible since the initial coverage is already close to the optimal diagonal and therefore does not leave much space for improvement. Comparing the approach with and without regularization, the coverage appears clearly improved by regularization, especially for the smaller training budgets. This signals a great success for the regularized approach, if one relies only on the *classical coverage*.

To investigate the performance of these models more deeply, we utilize the reference posterior, which is provided by the `sbibm` package for this benchmark task. For a fixed measurement $\hat{x}$, $10\,000$ samples are drawn from estimated posterior $q(\theta|\hat{x})$ and the true posterior $p(\theta|\hat{x})$, respectively. Based on those samples, Figure 4 shows the parameter regions with the highest probability of the true (orange) and estimated (blue) probability density corresponding to the same models shown in the previous figure.

The models without regularization show the expected trend of better agreement between the true and approximated distribution for larger simulation budgets. However, the regularized models show no improvement for larger training sizes and, in general, have less conformance with the true posterior. This is especially visible for the diagonal plots in Figure 4. The models without regularization perform better than the models with regularization, which confirms that the regularizer harms the optimization (see Section 4). This stands in opposition to the result of the *classical coverage* performed before.

This apparent contradiction corresponds to the general blind spot of the *classical coverage*, which allows for two different distributions to achieve perfect coverage (compare Example 3.3). The model distribution learned by the regularized model, resembles the prior very closely, which has been identified in the literature, e.g. Delaunoy et al. (2023), and can also be shown in the simple 1D case (Figure 8). With this experiment, we show that this mathematical case occurs in practice, specifically when using a training method that optimizes for perfect classical coverage. Hence, we show empirically that distributions close to the prior also have perfect *classical coverage*. In this example, we are only able to detect this by using true posterior samples, which is however not possible in practical applications.

## 6.2 The Ratio Coverage

Therefore, we propose the *ratio coverage*, which can discriminate these models correctly without using samples from the true posterior. In Figure 5 the *ratio coverage* plots based on the trained ratio estimation are compared with the *classical coverage* plots, for the same models. The models themselves are identically and have not been retrained, hence the *expected conditional classical coverage* is identical to Figure 3.

The *expected conditional ratio coverage* plots also show an improvement when comparing the step from low to high simulation budget. But in contrast to the classical coverage, the ratio coverage favors the unregularized over the regularized model for all simulation budgets. The same conclusion follows from the values of the *total variation norm*, calculated from the same trained ration estimation. For comparison, also the Kullback-Leibler divergence is calculated using the ratio estimation and supports previous findings. Specifically, using the TV regularizer hinders the performance of the models.

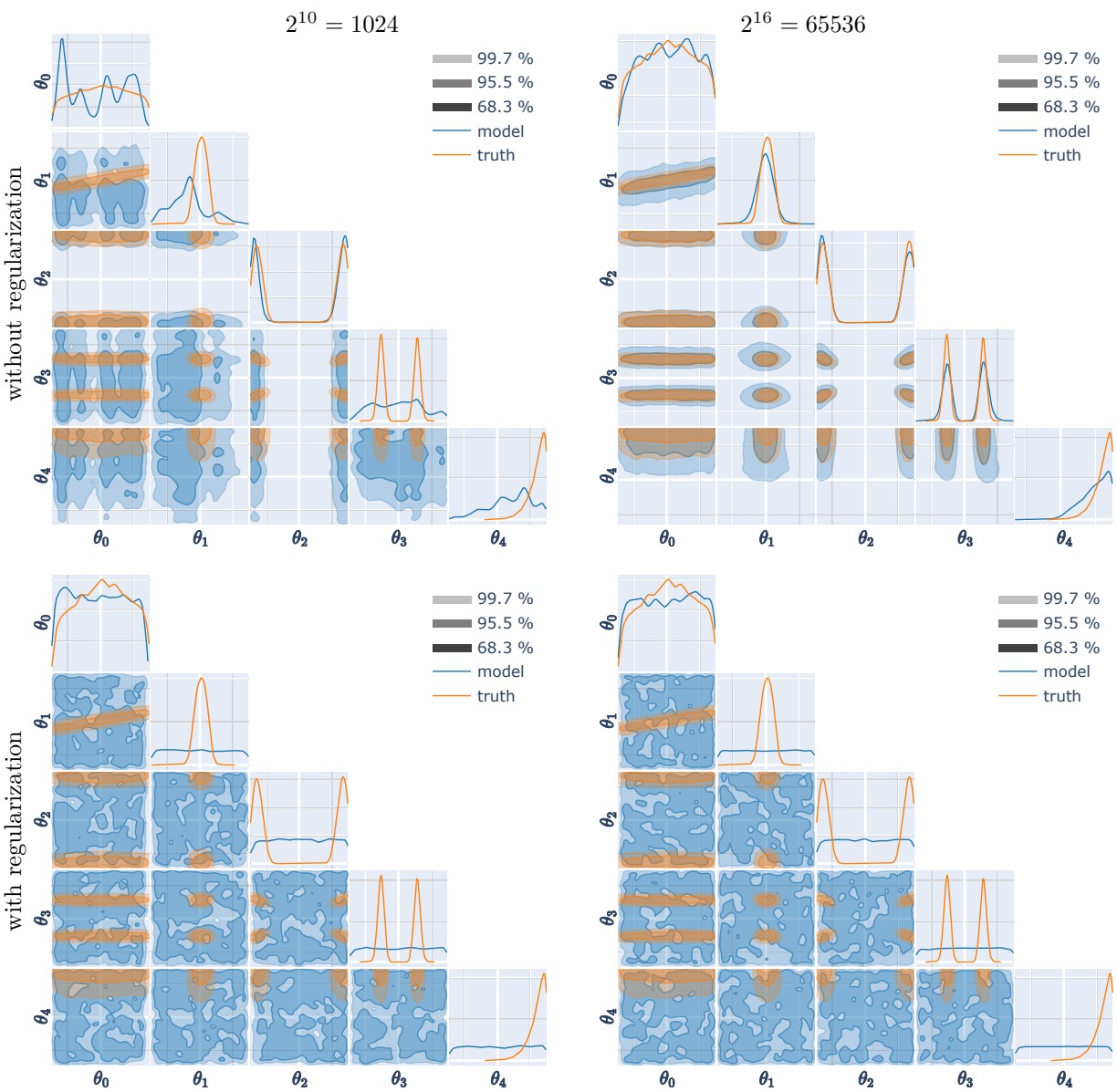

Figure 4: These plots reveal the devastating effect of the TV regularization despite its apparent positive effect on the *classical coverage*. They show the true and model posterior distribution for the SLCP benchmark task with simulation budgets $2^{10}, 2^{16}$ with and without TV regularization as contour plot of the three specified probability levels. The distributions are shown for the same fixed observation $\hat{x}$, therefore only the 5 dimensional $\theta$ dependence is depicted.

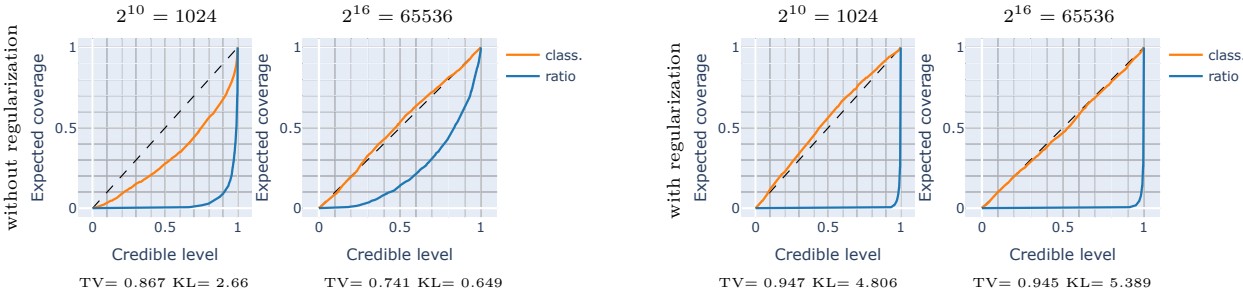

Figure 5: In contrast to the *classical coverage*, the *ratio coverage* can detect negative effect of the TV regularization and favors the unregularized models. These plots show the direct comparison between the classical and ratio coverage based on the same models shown previously for simulation budgets $2^{10}, 2^{16}$ with and without TV regularization. The coverage's are determined using Algorithm 2 ($M = N = 1024$). The TV and KL norm is calculated over the same data.

Hence, we have shown that also in practice the *ratio coverage* has improved discriminating capabilities compared to the *classical coverage* and the TV norm can be evaluated as a single value performance metric. Additionally, this performance metric offers the same benefits as the *classical coverage*, easy to interpret, easy to use without access to the true posterior samples. But there is the additional computational cost of learning the ratio from a set of samples.

## 6.3 Classical Coverage Regularization

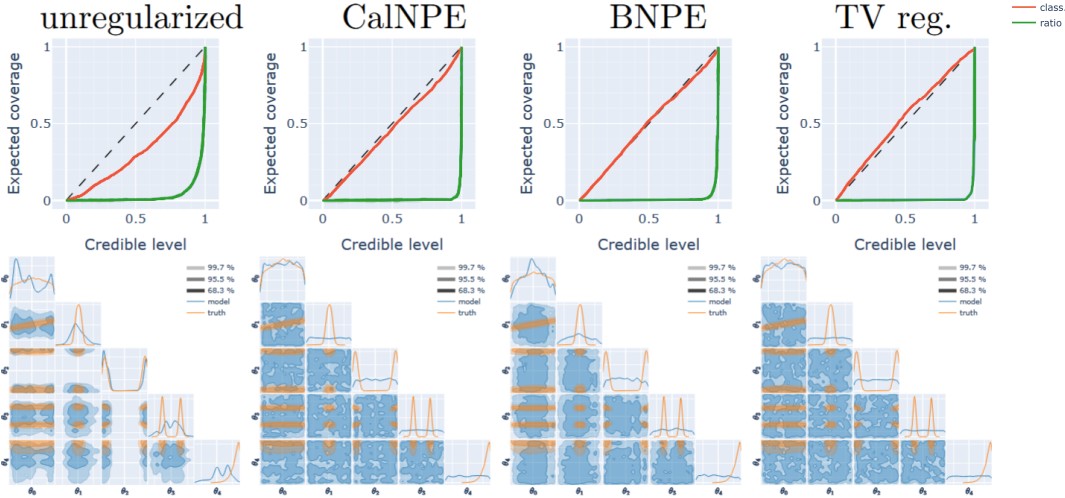

Figure 6: The *classical* (red line) and *ratio coverage* (green line) are shown for the SLCP task with a simulation budget of $2^{10}$ for standard NPE (KL divergence), CalNPE (Falkiewicz et al. (2023)), BNPE (Delaunoy et al. (2023)) and TV (Section 5) regularized models using Algorithm 2 ($M = N = 1024$).

This section extends the previously shown blind spot problem of the *classical coverage* using other regularization techniques from the literature. In accordance with Figures 4 and 5, Figure 6 additionally shows models trained using the CalNPE (Falkiewicz et al. (2023)) and BNPE (Delaunoy et al. (2023)) method. All presented regularized models show the blind spot behavior where the *classical coverage* plot gives ideal results while the posterior plot clearly deviates from the ground truth. In contrast, the *ratio coverage* favors the unregularized model, which is in agreement with the observation of the distribution plots.

This underlines the problem of the lacking discriminating power of the *classical coverage* in scenarios where the true posterior distribution is not available and a direct comparison between the learned and true posterior is

therefore impossible. Additionally, it shows that it is not a problem of the specific regularization technique, but is related to the general blind spot problem of the *classical coverage*, which disqualifies it as a regularization objective.

### 6.4 Unconditional Coverage

Lastly, we compare the *unconditional* with the *expected conditional coverage* plot from the previous section. Figure 7 visualizes the same models as Figure 5 using the *classical* and *ratio coverage*, but calculates the plots using the *unconditional coverage plots*. Although this method is mathematically and computationally simpler, the plots are very similar and show overall the same features. Although these results look very promising, a limitation of this example is small number of parameter. The joint parameter space of `SLCP` only consist of 13 parameter and going to higher dimensional problems might not work so well. In Section A.4 in the Appendix, we further compare both methods and also investigate their dependents on the number of samples, i.e. parameters $N$ and $M$).

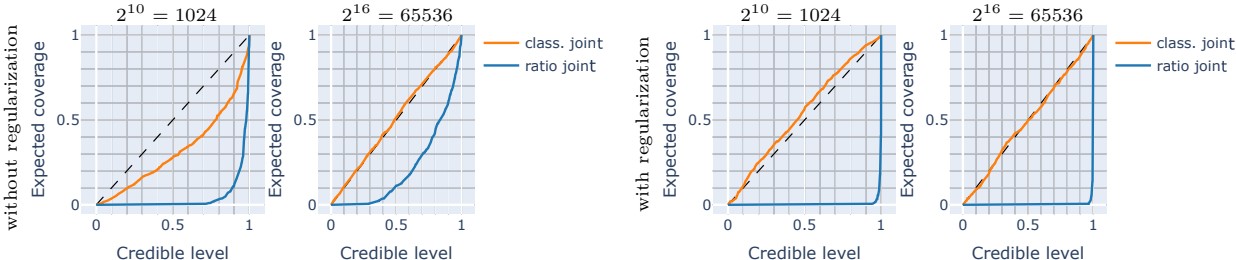

Figure 7: Using the *unconditional coverage* plot instead of the *expected conditional coverage* plot reduces the number of required samples, is mathematically simpler and reproduces identical results for *classical* and *ratio coverage*. Reproducing Figure 5 for simulation budgets $2^{10}, 2^{16}$ with and without TV regularization, but using the *unconditional coverage* plot (Algorithm 1 with $N = M = 1024$).

## 7 Conclusion

The motivation for this work originates from the blind spots of the *classical coverage* and its limited explanatory power. Specifically training objectives tuned to improve the coverage can very easily work in the opposite way and hinder the successfully training. Additionally in these situations, the *expected conditional classical coverage* plot is not able to detect the failure but instead signals a perfect performance.

We develop an adversary total variation objective function, derived from base principles, to optimize for the *classical coverage*. Although the expected coverage can be substantially improved, we provide empirical evidence and theoretical arguments that the quality of the surrogate model has actually been decreased. Since this deterioration remains undetected by the *classical coverage*, we propose the *ratio coverage* with increased discriminatory capabilities. At the same time, the *ratio coverage* is easy to use and interpret and does not require access to the true posterior. Lastly, we propose the *unconditional coverage* as an alternative to the *expected conditional coverage*, which reduces the computational cost of the calculation while also being mathematical more intuitive. To summarize, we showed that the KL divergence optimizes for the coverage without additional objective, instead we encourage to improve diagnostics by using the ratio coverage.

### Broader Impact Statement

The implications of SBI are similar to other SBI methods and are primarily scientific, but one must be careful to use an accurate generative model and to carefully test results. We emphasize the importance of the diagnostics in our paper, because untested inference can lead to incorrect conclusions, which can be missed by practitioners doing inference in any field. Special care applies to results that may influence human behavior or factors responsible for climate change.

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

# A  Appendix

## A.1  Proofs and further Results

**Definition A.1** (Kolmogorov–Smirnov divergence)**.** *We also highlight the* Kolmogorov–Smirnov divergence *between probability measures $P$ and $Q$ on $\bar{\mathbb{R}} = \mathbb{R} \,\dot{\cup}\, \{\pm\infty\}$, which is described by:*[2]

$$\mathrm{KS}(P\|Q) := \sup_{t\in\bar{\mathbb{R}}} |P([t,\infty]) - Q([t,\infty])|. \tag{38}$$

---

[2]Note that, for the definition of the Kolmogorov-Smirnov divergence, we took the intervals $[t,\infty]$ instead of (the equivalent choice of) $[-\infty, t]$ or corresponding (half-)open intervals, which can also be found in the literature.

**Remark A.2.** *Because we assumed that the marginals agree: $P(X) = Q(X)$, we also have the following tighter inequalities:*

$$\bar{\Delta}_g(P\|Q) = \sup_{\gamma \in [0,1]} |P(\mathcal{C}_g(\gamma)) - Q(\mathcal{C}_g(\gamma))| \tag{39}$$

$$\leq \sup_{\gamma \in [0,1]} \mathbb{E}_{P(X)} \left[ \left| P(\Theta \in \mathcal{C}_g^X(\gamma)|X) - Q(\Theta \in \mathcal{C}_g^X(\gamma)|X) \right| \right] \tag{40}$$

$$\leq \mathbb{E}_{P(X)} \left[ \sup_{\gamma \in [0,1]} \left| P(\Theta \in \mathcal{C}_g^X(\gamma)|X) - Q(\Theta \in \mathcal{C}_g^X(\gamma)|X) \right| \right] \tag{41}$$

$$= \mathbb{E}_{P(X)} \left[ \sup_{\gamma \in [0,1]} \left| P(g(\Theta, X) \geq t_\gamma(X)|X) - Q(g(\Theta, X) \geq t_\gamma(X)|X) \right| \right] \tag{42}$$

$$\leq \mathbb{E}_{P(X)} \left[ \sup_{t \in \mathbb{R}} |P(g(\Theta, X) \geq t|X) - Q(g(\Theta, X) \geq t|X)| \right] \tag{43}$$

$$= \mathbb{E}_{P(X)} \left[ \mathrm{KS}(g_\#^X P(\Theta|X) \| g_\#^X Q(\Theta|X)) \right] \tag{44}$$

$$\leq \mathbb{E}_{P(X)} \left[ \mathrm{TV}(g_\#^X P(\Theta|X) \| g_\#^X Q(\Theta|X)) \right] \tag{45}$$

$$= \mathrm{TV} \left( (g_\#^X P(\Theta|X)) \otimes P(X) \| (g_\#^X Q(\Theta|X)) \otimes P(X) \right) \tag{46}$$

$$= \frac{1}{2} \sup_{\substack{f:\, \bar{\mathbb{R}} \times \mathcal{X} \to [-1,1] \\ measurable}} |\mathbb{E}_P[f(g(\Theta, X), X)] - \mathbb{E}_Q[f(g(\Theta, X), X)]|, \tag{47}$$

*where for $x \in \mathcal{X}$ we abbreviate the partially evaluated map:*

$$g^x:\ \Theta \to \bar{\mathbb{R}}, \qquad\qquad g^x(\theta) := g(\theta, x). \tag{48}$$

**Remark A.3.**

1. *Note that the supremum in Equation (27) is by Theorem 2.3 attained for:*

$$f^\star(y, x) := \mathrm{sgn} \log \frac{q(y|x)}{p(y|x)}, \tag{49}$$

   *where $q(y|x)$ and $p(y|x)$ denote the probability densities of the push-forward probability measures $g_\#^x Q(\Theta|X = x)$ and $g_\#^x P(\Theta|X = x)$, resp., on $\bar{\mathbb{R}}$.*

2. *If we choose $g$ either to be:*

$$g(\theta, x) = \log \frac{q(\theta|x)}{p(\theta|x)} \qquad or \qquad g(\theta, x) = \mathrm{sgn} \log \frac{q(\theta|x)}{p(\theta|x)}, \tag{50}$$

   *then by Theorem 4.1 the bound in Equation (27) equals:*

$$\mathbb{E}_{P(X)} \left[ \mathrm{TV}(g_\#^X P(\Theta|X) \| g_\#^X Q(\Theta|X)) \right] = \mathbb{E}_{P(X)} \left[ \mathrm{TV}(P(\Theta|X) \| Q(\Theta|X)) \right] \tag{51}$$

$$= \mathrm{TV}(P\|Q) \tag{52}$$

$$= \frac{1}{2} \sup_{\substack{f:\, \Theta \times \mathcal{X} \to [-1,1] \\ measurable}} |\mathbb{E}_P[f] - \mathbb{E}_Q[f]|, \tag{53}$$

   *and the supremum is by Theorem 2.3 attained with $f^\star$ given by:*

$$f^\star(\theta, x) = \mathrm{sgn} \log \frac{q(\theta, x)}{p(\theta, x)} = \mathrm{sgn} \log \frac{q(\theta|x)}{p(\theta|x)}. \tag{54}$$

**Proposition A.4.** *Let the notation be like in Definition 3.1.*

1. *We can rewrite the occurring quantities with help of the push-forward measures $g_\# P$ and $g_\# Q$ from Equation 7 as follows:*

$$P(\mathcal{C}_g(t)) = (g_\# P)([t, \infty]), \qquad\qquad Q(\mathcal{C}_g(t)) = (g_\# Q)([t, \infty]). \tag{55}$$

2. *With this we see that the coverage gap between $P$ and $Q$ w.r.t. $g$ can be identified with the* Kolmogorov-Smirnov divergence *between their corresponding push-forward probability measures:*

$$\Delta_g(P\|Q) = \sup_{t \in \bar{\mathbb{R}}} |(g_\# P)([t, \infty]) - (g_\# Q)([t, \infty])| = \mathrm{KS}(g_\# P \| g_\# Q). \tag{56}$$

3. *We then directly get the following upper bound with the* total variation norm *(see Remark A.5):*

$$\mathrm{KS}(g_\# P \| g_\# Q) \leq \sup_{B \in \mathcal{B}_{\bar{\mathbb{R}}}} |(g_\# P)(B) - (g_\# Q)(B)| = \mathrm{TV}(g_\# P \| g_\# Q) \tag{57}$$

$$= \frac{1}{2} \sup_{\substack{f: \bar{\mathbb{R}} \to [-1,1] \\ measurable}} |\mathbb{E}_P[f \circ g] - \mathbb{E}_Q[f \circ g]|, \tag{58}$$

*where the first supremum ranges over all measurable subsets $B$ of $\bar{\mathbb{R}}$, and, the second supremum over all measurable maps $f$ from $\bar{\mathbb{R}}$ to $[-1, 1]$.*

**Remark A.5.** *We have the following well-known inequalities:*

1. $\mathrm{KS}(P\|Q) \leq \mathrm{TV}(P\|Q)$ *Kelbert (2023).*

2. *Pinsker-Bretagnolle-Huber inequality (see Bretagnolle & Huber (1978)):*

$$\mathrm{TV}(P\|Q)^2 \leq \min\left(\frac{1}{2}\mathrm{KL}(P\|Q), 1 - \exp\left(-\mathrm{KL}(P\|Q)\right)\right). \tag{59}$$

**Theorem A.6** (See Iosif Pinelis (2023)). *We have the following identities:*

$$2 \cdot \mathrm{TV}(P\|Q) = \int_{\mathcal{Z}} |p - q| \, d\nu = \sup_{\substack{f: \mathcal{Z} \to [-1,1] \\ measurable}} |\mathbb{E}_P[f] - \mathbb{E}_Q[f]|, \tag{60}$$

*and the supremum on the rhs is attained for:*

$$f := \mathbb{1}_{\mathcal{Z}_>} - \mathbb{1}_{\mathcal{Z}_<} = \mathrm{sgn}\log\left(\frac{p}{q}\right), \tag{61}$$

*where*

$$\mathcal{Z}_> := \{z \in \mathcal{Z} \,|\, p(z) > q(z)\} = \left\{z \in \mathcal{Z} \,\middle|\, \log\frac{p(z)}{q(z)} > 0\right\} \tag{62}$$

$$\mathcal{Z}_< := \{z \in \mathcal{Z} \,|\, p(z) < q(z)\} = \left\{z \in \mathcal{Z} \,\middle|\, \log\frac{p(z)}{q(z)} < 0\right\}, \tag{63}$$

*where $p$ and $q$ are densities for $P$ and $Q$, resp., w.r.t. any fixed joint dominating measure $\nu$, e.g. $\nu = P + Q$.*

*Proof.* For a measurable subset $A \subseteq \mathcal{Z}$ we get:

$$|P(A) - Q(A)| = \left| \int_A p(z) \, \nu(dz) - \int_A q(z) \, \nu(dz) \right| \tag{64}$$

$$= \left| \int_A (p(z) - q(z)) \, \nu(dz) \right| \tag{65}$$

$$= \left| \underbrace{\int_{A_>} (p(z) - q(z)) \, \nu(dz)}_{\geq 0} - \underbrace{\int_{A_<} (q(z) - p(z)) \, \nu(dz)}_{\geq 0} \right| \tag{66}$$

$$\leq \max \left( \int_{A_>} (p(z) - q(z)) \, \nu(dz), \int_{A_<} (q(z) - p(z)) \, \nu(dz) \right) \tag{67}$$

$$\leq \max \left( \int_{\mathcal{Z}_>} (p(z) - q(z)) \, \nu(dz), \int_{\mathcal{Z}_<} (q(z) - p(z)) \, \nu(dz) \right). \tag{68}$$

Note that the maximum on the rhs can be achieved by either putting $A = \mathcal{Z}_>$ or $A = \mathcal{Z}_<$, depending on which set maximizes the rhs. Also note that for $A = \mathcal{Z}$ we have $P(\mathcal{Z}) = 1 = Q(\mathcal{Z})$, so the lhs vanishes. The above equality then shows:

$$\int_{\mathcal{Z}_>} (p(z) - q(z)) \, \nu(dz) = \int_{\mathcal{Z}_<} (q(z) - p(z)) \, \nu(dz). \tag{69}$$

So both values in the maximum above are the same and the maximium can thus also be written as the convex combination:

$$\mathrm{TV}(P\|Q) = \max \left( \int_{\mathcal{Z}_>} (p(z) - q(z)) \, \nu(dz), \int_{\mathcal{Z}_<} (q(z) - p(z)) \, \nu(dz) \right) \tag{70}$$

$$= \frac{1}{2} \left( \int_{\mathcal{Z}_>} (p(z) - q(z)) \, \nu(dz) + \int_{\mathcal{Z}_<} (q(z) - p(z)) \, \nu(dz) \right) \tag{71}$$

$$= \frac{1}{2} \left( \int f(z) \cdot p(z) \, \nu(dz) - \int f(z) \cdot q(z) \, \nu(dz) \right) \tag{72}$$

$$= \frac{1}{2} \left( \mathbb{E}_P[f] - \mathbb{E}_Q[f] \right), \tag{73}$$

where:

$$f := \mathbb{1}_{\mathcal{Z}_>} - \mathbb{1}_{\mathcal{Z}_<} = \mathrm{sgn} \log \left( \frac{p}{q} \right). \tag{74}$$

Note that we can also write:

$$\mathrm{TV}(P\|Q) = \max \left( \int_{\mathcal{Z}_>} (p(z) - q(z)) \, \nu(dz), \int_{\mathcal{Z}_<} (q(z) - p(z)) \, \nu(dz) \right) \tag{75}$$

$$= \frac{1}{2} \left( \int_{\mathcal{Z}_>} (p(z) - q(z)) \, \nu(dz) + \int_{\mathcal{Z}_<} (q(z) - p(z)) \, \nu(dz) \right) \tag{76}$$

$$= \frac{1}{2} \left( \int_{\mathcal{Z}_>} |p(z) - q(z)| \, \nu(dz) + \int_{\mathcal{Z}_<} |p(z) - q(z)| \, \nu(dz) \right) \tag{77}$$

$$= \frac{1}{2} \int_{\mathcal{Z}} |p(z) - q(z)| \, \nu(dz). \tag{78}$$

Now if $h: \mathcal{Z} \to [-1, 1]$ is any measurable map, we get:

$$|\mathbb{E}_P[h] - \mathbb{E}_Q[h]| = \left| \int h(z) \cdot p(z) \, \nu(dz) - \int h(z) \cdot q(z) \, \nu(dz) \right| \tag{79}$$

$$= \left| \int h(z) \cdot (p(z) - q(z)) \, \nu(dz) \right| \tag{80}$$

$$\leq \int \underbrace{|h(z)|}_{\leq 1} \cdot |p(z) - q(z)| \, \nu(dz) \tag{81}$$

$$\leq \int |p(z) - q(z)| \, \nu(dz) \tag{82}$$

$$= 2 \cdot \mathrm{TV}(P\|Q). \tag{83}$$

So we finally get:

$$\mathrm{TV}(P\|Q) = \frac{1}{2} \sup_{\substack{h:\mathcal{Z} \to [-1,1] \\ \text{measurable}}} |\mathbb{E}_P[h] - \mathbb{E}_Q[h]|. \tag{84}$$

So all claims are shown. $\qquad\square$

**Theorem A.7.** *We now consider the functions given by the log-ratio between the corresponding probability densities and its sign function:*

$$g_{\mathrm{lr}}(z) := \log \frac{q(z)}{p(z)}, \qquad\qquad g_s(z) := \mathrm{sgn} \log \frac{q(z)}{p(z)}. \tag{85}$$

*Both corresponding coverage plots $\Gamma_g(P\|Q)$ are able to distinguish $P$ and $Q$. Furthermore, we have for those two $g$'s the identity:*

$$\mathrm{TV}(g_\# P \| g_\# Q) = \mathrm{TV}(P\|Q). \tag{86}$$

*Proof.* We first show the last claim. It is clear that:

$$\mathrm{TV}(g_\# P \| g_\# Q) \leq \mathrm{TV}(P\|Q). \tag{87}$$

For the reverse, note that by Theorem 2.3 we have with $g(z) = \mathrm{sgn} \log \frac{q(z)}{p(z)}$:

$$\mathrm{TV}(P\|Q) = \frac{1}{2} |\mathbb{E}_{Z\sim P}[g(Z)] - \mathbb{E}_{Z\sim Q}[g(Z)]| \tag{88}$$

$$\leq \frac{1}{2} \sup_{h:\mathbb{R} \to [-1,1]} |\mathbb{E}_{Z\sim P}[h \circ g(Z)] - \mathbb{E}_{Z\sim Q}[h \circ g(Z)]| \tag{89}$$

$$\leq \frac{1}{2} \sup_{h:\mathbb{R} \to [-1,1]} |\mathbb{E}_{Y\sim(g_\# P)}[h(Y)] - \mathbb{E}_{Y\sim(g_\# Q)}[h(Y)]| \tag{90}$$

$$= \mathrm{TV}(g_\# P \| g_\# Q). \tag{91}$$

This shows the claim. Similarly, for $g(z) = \log \frac{q(z)}{p(z)}$.

Now assume that:

$$0 = \Delta_g(P\|Q) = \mathrm{KS}(g_\# P \| g_\# Q), \tag{92}$$

then $g_\# P = g_\# Q$ as KS is a proper divergence. In particular, we get:

$$0 = \mathrm{TV}(g_\# P \| g_\# Q) = \mathrm{TV}(P\|Q), \tag{93}$$

which implies $P = Q$, as TV is a proper divergence. $\qquad\square$

Theorem 4.1 together with Equation (25) now show the following:

**Corollary A.8.** *Assume that the marginals are equal: $P(X) = Q(X)$, and, that we use the log ratio $g(\theta, x) = \log \frac{q(\theta|x)}{p(\theta|x)}$. Then we get the following inequalities:*

$$\bar{\Delta}_g(P\|Q) \leq \mathbb{E}_{P(X)}\left[\mathrm{KS}(g_\#^X P(\Theta|X)\|g_\#^X Q(\Theta|X))\right] \tag{94}$$

$$\leq \mathbb{E}_{P(X)}\left[\mathrm{TV}(g_\#^X P(\Theta|X)\|g_\#^X Q(\Theta|X))\right] \tag{95}$$

$$= \mathbb{E}_{P(X)}\left[\mathrm{TV}(P(\Theta|X)\|Q(\Theta|X))\right] \tag{96}$$

$$= \mathrm{TV}(P\|Q). \tag{97}$$

*Furthermore, if $\bar{\Delta}_g(P\|Q) = 0$ then we have: $P = Q$.*

*Proof.* The chain of inequalities follows from Theorem 4.1 together with Equation (25).

If now $\bar{\Delta}_g(P\|Q) = 0$, then also $\mathbb{E}_{P(X)}\left[\mathrm{KS}(g_\#^X P(\Theta|X)\|g_\#^X Q(\Theta|X))\right] = 0$. Since KS is a proper divergence, we have that for $P(X)$-almost-all $x \in \mathcal{X}$ the equality:

$$g_\#^x P(\Theta|X = x) = g_\#^x Q(\Theta|x), \tag{98}$$

which implies:

$$\left[\mathrm{TV}(g_\#^x P(\Theta|X = x)\|g_\#^x Q(\Theta|X = x))\right] = 0. \tag{99}$$

By the above identities we thus get:

$$0 = \mathbb{E}_{P(X)}\left[\mathrm{TV}(g_\#^X P(\Theta|X)\|g_\#^X Q(\Theta|X))\right] = \mathrm{TV}(P\|Q). \tag{100}$$

Since TV is a proper divergence, this implies $P = Q$. $\square$

**Example A.9** (Superlevel sets cannot discriminate probability measures)**.** *Let $q$ be a probability density supported on $[-2, -1]$ and $p$ be the probability density given by:*

$$p(z) := \frac{1}{2}q(z) + \frac{1}{2}q(z - 3), \tag{101}$$

*which is supported on $[-2, -1] \,\dot\cup\, [1, 2]$. Let $Q$ and $P$ be the corresponding probability measures of $\mathbb{R}$ w.r.t. $q$ and $p$, resp. Now consider the superlevel set for $p$ for thresholds $t > 0$:*

$$\mathcal{C}_p(t) := \{z \in \mathbb{R} \,|\, p(z) \geq t\} \tag{102}$$

$$= \{z \in [-2, -1] \,|\, p(z) \geq t\} \,\dot\cup\, \{z \in [1, 2] \,|\, p(z) \geq t\} \tag{103}$$

$$= \left\{z \in [-2, -1] \,\middle|\, \frac{1}{2}q(z) \geq t\right\} \,\dot\cup\, \left\{z \in [1, 2] \,\middle|\, \frac{1}{2}q(z - 3) \geq t\right\} \tag{104}$$

$$= q^{-1}([2t, \infty)) \,\dot\cup\, \left(q^{-1}([2t, \infty)) + 3\right) \tag{105}$$

$$=: I_1 + I_2, \tag{106}$$

*with $I_1 := q^{-1}([2t, \infty)) \subseteq [-2, -1]$ and $I_2 := I_1 + 3 \subseteq [1, 2]$. With this we get:*

$$P(\mathcal{C}_p(t)) = P(I_1) + P(I_2) \tag{107}$$

$$= \int_{I_1} p(z)\, dz + \int_{I_1+3} p(z)\, dz \tag{108}$$

$$= \int_{I_1} \left( \frac{1}{2} q(z) + \frac{1}{2} \underbrace{q(z-3)}_{=0 \ on \ I_1} \right) dz + \int_{I_2} \left( \frac{1}{2} \underbrace{q(z)}_{=0 \ on \ I_2} + \frac{1}{2} q(z-3) \right) dz \tag{109}$$

$$= \frac{1}{2} \left( \int_{I_1} q(z)\, dz + \int_{I_1+3} q(\underbrace{z-3}_{=:z'})\, dz \right) \tag{110}$$

$$= \frac{1}{2} \left( \int_{I_1} q(z)\, dz + \int_{I_1} q(z')\, dz' \right) \tag{111}$$

$$= \int_{I_1} q(z)\, dz \tag{112}$$

$$= Q(I_1) \tag{113}$$

$$= Q(I_1) + \overbrace{Q(I_2)}^{=0} \tag{114}$$

$$= Q(\mathcal{C}_p(t)). \tag{115}$$

*This shows that $P$ and $Q$ agree on all superlevel sets of $P$, but clearly $P \neq Q$. Note that for $t = 0$ we have $\mathcal{C}_p(t) = \mathbb{R}$ and $P(\mathcal{C}_p(t)) = 1 = Q(\mathcal{C}_p(t))$ as well.*

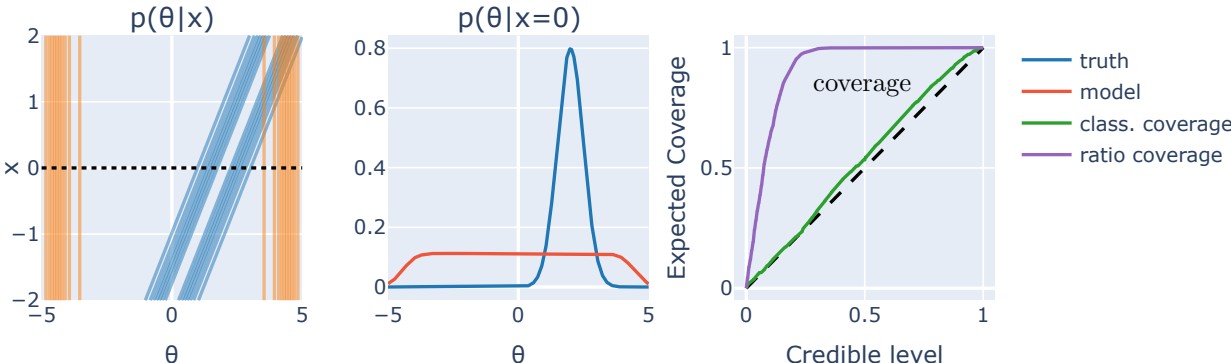

Figure 8: Analog to Figure 2, this example extends the same scenario but uses 10 peaks in the model distribution instead of two. As such, the model distribution becomes quite flat and resembles what we see in the contour plots in the experiment. The true and model dist. given by $p(\theta|x)$ and $q(\theta|x)$ are shown in the first image and the cross section for $x = 0$ is presented in the second plot (black dotted line). The model distribution does not depend on $x$ in contrast to the previous figure and the 10 peaks are equally distributed between $-4$ and $4$. The third plot show the approximated classical and ratio coverage based on the Algorithm 2 for a finite data sets drawn from the distributions.

## A.2   Setup

In our experiments, we follow the procedure from Hermans et al. (2022) for evaluating SBI models and also use the implementations from the `sbi 0.21.0` package (Tejero-Cantero et al. (2020)), in order to be comparable to literature. All calculations are performed on a Windows Server 2019 with a AMD Ryzen Threadripper 3970X processor with 32 cores (3693 MHz, 64 logical processors) and NVIDIA GeForce RTX 2070 SUPER. The total available physical memory is 128 GB (3200 MHz).

We conducted two kinds of experiments. First, using the the `NPE` class from `sbi` package, we use normalizing flow networks together with the standard Kullback-Leibler training objective to create a base line results close to the literature reference. We chose different simulation budgets $2^{10}, 2^{16}$ and train each model for 50 epochs. After each epoch, the validation set of $2^{11}$ elements is evaluated and the model with the lowest validation loss is restored after the training. We observe similar results for different batches sizes, if the learning rate is scaled accordingly, and therefore chose a batch size of 512 for maximal efficiency. The data is simulated using the standard benchmark `SLCP` implemented in the `sbibm 1.1.0` package (Lueckmann et al. (2021)). It simulates a fictive problem with 5 parameters and 8 observables sampled from a multivariate Gaussian whose mean and covariance matrix are parameterized.

Second, the TV regularized case is trained with the additional adversary optimization performed on the same data as the training of the surrogate model. To ensure good comparability with the reference models, the starting conditions are the same with and without regularization. Custom implementations were only necessary for the adversary training part of the TV regularizer using the `PyTorch 2.2.0` (Paszke et al. (2019)) and `LAMPE 0.8.2` (Rozet et al. (2021)) package. The adversary network consists of a dense neuronal network with ReLU activation functions and a *tanh* final layer. It receives 10 update steps for each update to the surrogate model, to ensure good convergence. The regularization coefficient (Equation (35)) was set to $\lambda = 10^2$, which roughly aligns both loss contribution in strength at the beginning of the training.

For the evaluation of the *ratio coverage*, another classifier is trained. It uses an AdamW optimizer with a batch size of 512 and a dense network with $5 \times 64$ hidden features and ELU activation trained for 32 epochs. An architecture search was performed to find these values. However, for our experiments, the training of the NPE model far surpassed the training of the classifier in terms of computational cost and amount of required training data. Hence, we selected a classifier architecture that is sufficiently flexible, but did not optimize it extensively. It is trained using a binary cross entropy loss on a total of 1000 data points consisting of $(\theta, x)$ and $(\hat{\theta}, x)$ with $\theta$ being generated from the simulator $\theta \sim p(\theta|x)$ and $\hat{\theta}$ from the surrogate model $\hat{\theta} \sim q(\hat{\theta}|x)$. Then the optimal classifier is the probability ratio as discussed in Theorem 4.1.

The main challenge when using the TV objective function is the adversary training, which is unstable. Using the hyper-parameter $\lambda$, it can be controlled at the cost of reducing the regularization effect.

All code to reproduce the results presented in this paper is available: https://anonymous.4open.science/r/ratio_coverage-E777/.

## A.3 Two Moons

In the main manuscript we present results based on the `SLCP` benchmark test, that show improved discriminating power of the *ratio coverage*. In this section, we present analog findings based on the `two moons` benchmark test, which is lower dimensional compared to `SLCP`, but its shape is more difficult to learn.

Figure 9 and 10 confirm the findings of the main paper for the `two moons` task. First, Figure 9 clearly shows improved performance for the regularized based on the *classical coverage*, but Figure 10 shows that the performance should actually be worse. Hence, the same blind spot problem arises here as discussed for *SLCP*, but the *ratio coverage* was again not effected.

## A.4 Variance of Coverage plots

In this section we investigate the stability of the *expected conditional* and *unconditional coverage* plot (see Algorithm 2 and 1). The stability and computational cost of these coverage plots depends on the number of samples $n$ and $m$, where $n$ is related to the granularity of the coverage plot and $m$ is associated with the stability. In terms of computational simulation cost, the *expected conditional coverage* generates $n \times m$ simulations while the *unconditional coverage* only requires $n+m$ samples. To investigate the stability between both methods, we keep $n = 1024$ the same and consider $m = 128$ and $m = 512$.

In Figure 11 we show the standard deviation across 10 reevaluations of the *classical coverage* and *ratio coverage* using the *expected conditional* and *unconditional coverage* plot. The evaluated model was previously shown in Figure 5 and 7 in the main text. The *expected conditional* and *unconditional coverage* plot agree in

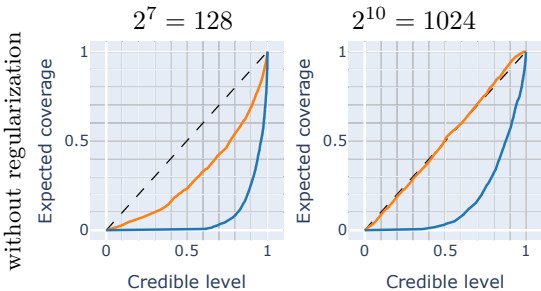 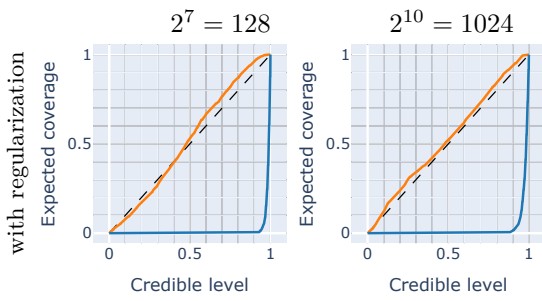

Figure 9: This figure compare the *classical coverage* (orange) and the *ratio coverage* (blue) for TV regularized and unregularized models trained with simulation budgets of $2^7, 2^{10}$. This plot is analog to Figure 5 in the main paper.

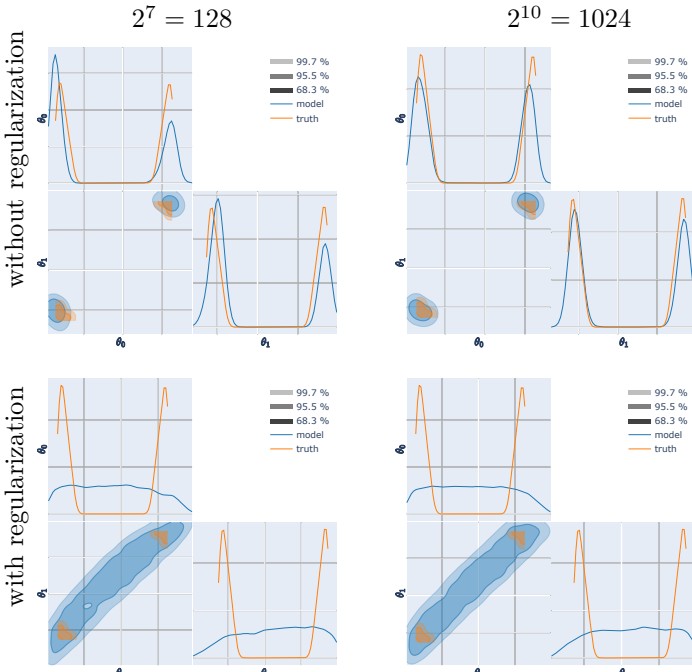

Figure 10: These plots reveal the devastating effect of the TV regularization despite its apparent positiv effect on the *classical coverage*. They show the true and model posterior distribution for the *two moons* benchmark task with simulation budgets $2^7, 2^{10}$ with and without TV regularization. This plot is analog to Figure 4 in the main paper.

all cases, which shows the *unconditional coverage* as a more cost efficient alternative. The standard deviation of both methods also roughly agrees, while the standard deviation is significantly decreased for the 512 sample case.

## A.5 Simple illustration for coverage

In this section we illustrate the interpretability of the *classical* and the *ratio coverage plot* expanding on the blind-spot problem shown in Figure 1. Fundamentally, the *coverage plot* is a statistical tool to measure the similarity between a true and a model distribution without direct access to the true distribution. However, for easier visualization we considering simple one-dimensional normal distributions $p$ and $q$ (shown in blue and red).

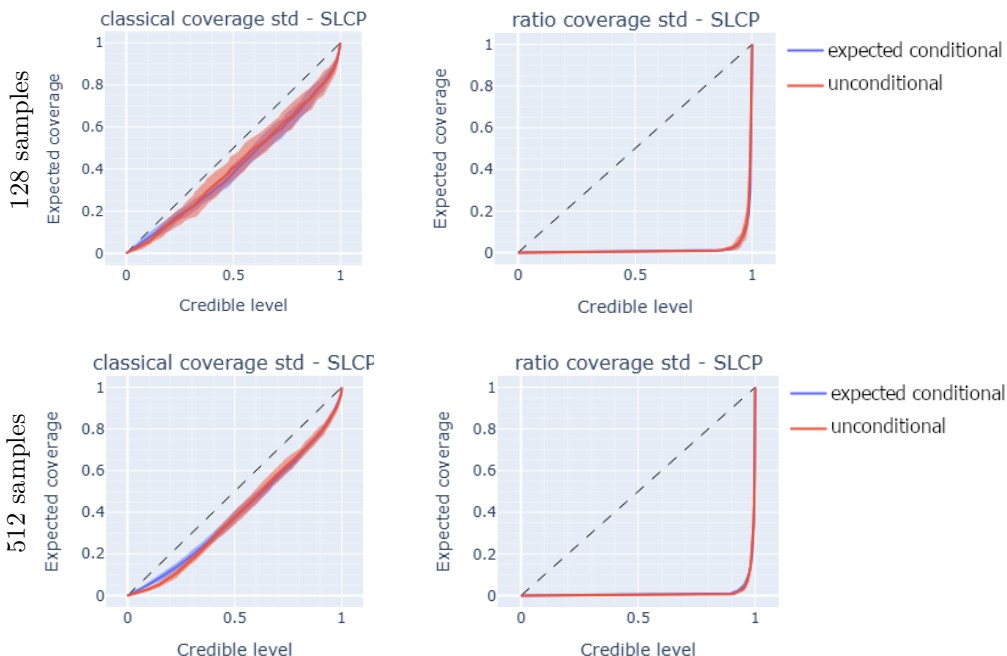

Figure 11: Plotting the standard deviation of 10 independent coverage plots of the unregularized models depicted in Figure 5 and Figure 7 with a budget of 1024 estimated over 128 and 512 test samples. The surrogate model has not been retrained. The blue line corresponds to the *expected conditional coverage* plot using Algorithm 2 and the red line corresponds to the *conditional coverage* plot using Algorithm 1.

Figure 12 consists of 5 sub-figures that each investigate a different true distribution, but have the same model distribution. The top, center figure considers the case where both distributions perfectly agree and we observe the expected perfect coverage from the *ratio* and the *classical coverage* plot. The second row considers a shift between both distributions and both coverage plots show a deformation below the optimal diagonal, while the *ratio coverage* exhibits higher sensitivity, i.e., more distance to the diagonal. The bottom row investigates a change in the width of the distribution and for the wider model distribution the *classical* and the *ratio coverage* show mirrored behavior. However, both coverages detect the disagreement between the distributions, because the distance from the diagonal is the similar, but the direction is different. Although, in these simple cases both coverages are able to detect the difference between the distributions, this does not hold in general, because of the blind spot of *ratio coverage* illustrated in Figure 1.

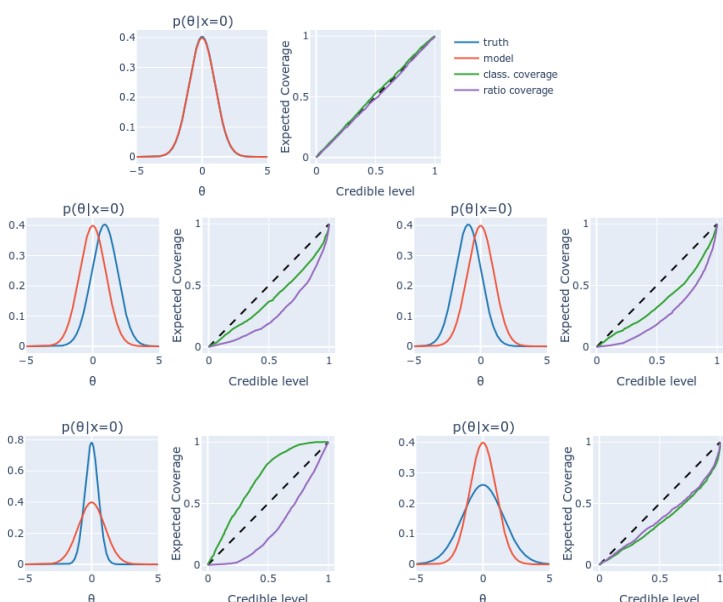

Figure 12: Plotting multiple simple cases to illustrate how the coverage behaves in comparison to the classical coverage plot. One can observe that for the left and right shifted case, the ratio coverage behaves similar to the classical coverage but is more sensitives. For the sharper and wider distribution example one can observe that the classical and ratio coverage agree almost perfect, except for the wider model where the ratio coverage gives the rotated result, which corresponds to the choice of the ration $\frac{p}{q}$ versus $\frac{q}{p}$.

