# OpenReview forum: "Generalizing Coverage Plots for Simulation-based Inference"
_TMLR — Accepted by TMLR_

### Review · Reviewer_vyzo · 2025-12-16

**Summary Of Contributions:**

The paper investigate methods of checking the correctness of approximations to the Bayesian posterior distribution based on "coverage" (also commonly known as calibration in the literature). Two types of coverage are given a rigorous and general definition. The authors investigate two particular versions: "classical coverage" and "ratio coverage". They also present an objective function based on an adversarial form of coverage. The adversarial approach is shown to improve classical coverage but nonetheless provide a poor fit to the posterior. Ratio coverage can detect this poor fit, and the authors also prove it has some more desirable theoretical properties: perfect coverage is attained if an only if the approximation equals the posterior.

## Strengths

* A coverage/calibration method avoiding several failure modes.

* Detailed proofs

* Examples clearly demonstrating main points of the paper.

## Weaknesses

* It would be good to further discuss "Discriminative Calibration: Check Bayesian Computation from Simulations and Flexible Classifier" by Yao and Domke (NeurIPS 2023). It seems possible to me that the proposed ratio coverage method is a special case of Yao and Domke's work.

* More details are needed on how the ratio coverage classifier is implemented, especially how training data is generated. (These details would have helped me check the previous point.)

* The paper only looks at two simple benchmark models.

* It would be good to give more details of computational cost.

* The paper has some grammar mistakes which occasionally make it hard to follow.

**Audience:**

Yes

**Audience Explanation:**

Section 4 proposes a novel adversarial regularisation method for NPE, but goes on to show (in Section 6) this has very poor performance. So this method seems of limited interest to readers. Perhaps the authors can add extra justification of why it's important.

The proposed ratio coverage method seems useful and of interest to TMLR's audience.

**Broader Impact Concerns:**

In my opinion, the paper's current Broader Impact Statement is sufficient.

**Claims And Evidence:**

Yes

**Claims Explanation:**

There are detailed and rigorous mathematical definitions and proofs. The examples are excellent, supporting the paper's claims about which methods work well or poorly.

**Requested Changes:**

## Critical

* Can you give more details on the computational cost of the proposed methods, especially training the ratio estimator.

* Further discuss "Discriminative Calibration: Check Bayesian Computation from Simulations and Flexible Classifier" by Yao and Domke (NeurIPS 2023). It seems possible to me that the proposed ratio coverage method is a special case of Yao and Domke's work.

* Give more details on how the ratio coverage classifier is implemented, especially how training data is generated. (These details would have helped me check the previous point.)

## Would strengthen work

* It would be ideal to investigate some more examples. But this is not absolutely necessary in my opinion.

* Discuss whether the method is only applicable to SBI, or also has applications to other approximate Bayesian methods.

## Minor

* Use a spelling and grammar checker to fix mistakes. These are mostly minor. An exception is the line before Definition 3.4, where there's a sentence fragment that I can't understand (referring to theorem 3.2, which doesn't appear in the paper).

* Many citations would be better in parentheses.

* Many equations are unnecessarily split over multiple lines, when they would fit on a single line.

* Comment on meaning of $\mathcal{B_Z}$. Is this an arbitrary $\sigma$-algebra? If it's meant to be a Borel $\sigma$-algebra then I think you need to assume something about the topology of $\mathcal{Z}$.

* On the line before equation (2), I think $\pm$ is a typo of $\pm \infty$.

* In Definition 3.4, do you need any extra assumptions to ensure $Q(A | X=x)$ is well defined?

* Page 6 says "we show that the expected conditional coverage gap... is bounded above by the total variation norm". Wasn't this already shown in Equation (19)?

* The paragraph before equation (25) mentions that regularity conditions are needed for this result. Can you expand on what these are, and give a citation for where a proof can be found.

* Page 6 says "...ratio coverage is based on the observation in Theorem A.4". Can you give more details, as the theorem statement is long and makes several observations.

* On page 7, the paragraph starting "We proofed" seems to repeat some material from Example 3.5. If this is intended, perhaps you could add a comment that you're recapping this.

* On page 7, is "expected unconditional coverage" a typo? Earlier I think you only define "unconditional coverage" and "expected conditional coverage".

* Appendices A.4 and A.5 should contain some text to explain why you include Figures 10 and 11.

---

> ### Author Response · Authors · 2026-01-06
> **Response to Reviewer vyzo**
>
> We thank the reviewer for their work and will address each of the concerns individually:
>
> ## Major
>
> 1. Can you give more details on the computational cost of the proposed methods, especially training the ratio estimator.
>
> _Throughout the paper, we present the unconditional coverage as an alternative to the expected conditional coverage (see Algorithm 2 and 1), which is the standard for the classical coverage in literature. However, our comparison shows that the unconditional coverage gives similarly good plots and reduces the computational cost of the evaluation, especially for expensive sampling methods. These results are discussed in section 5.3 Unconditional Coverage.
> Concerning the classifier, our method and the theory hold for general discriminating function, i.e., even if the ratio is not fitted perfectly, the ratio coverage has a high discriminating power. In all our test cases, a simple classifier and a short training were sufficient to achieve these results (see Appendix A.2 Setup) and other factors like sampling time took the majority of the evaluation time. The bottleneck, in terms of computational cost and performance, for simulation-based inference in our experiments was not the classifier, but the training of the NPE model itself. Other methods in the literature, like $\ell$-C2ST and DiscCali, have equal or even higher computational cost when training their classifiers._
>
> 2. Further discuss "Discriminative Calibration: Check Bayesian Computation from Simulations and Flexible Classifier" by Yao and Domke (NeurIPS 2023).
> It seems possible to me that the proposed ratio coverage method is a special case of Yao and Domke's work.
>
> _The generalized framework that is introduced in our manuscript is purposefully general. Its formulation allows for general discriminating functions $g$ such that we can compare different cases in one framework (e.g., classical coverage, ratio coverage, etc.). So, every other function $g$, however trained, can be used in this framework as well (with probably different properties).
> DiscCali also uses a learned test statistic, which does correctly detect the blind spot and agrees with the ratio coverage. However, DiscCali does not have the interpretability of a general coverage test (see Fig. 11) but estimates the value of the divergence. The paper presents different ways to train a classifier to estimate divergences based on the classifier two-sample test (C2ST) and proposes an algorithm to compute the p-value. It does not focus on coverage plots but on a label mapping for the training of the classifier._
>
> 3. Give more details on how the ratio coverage classifier is implemented, especially how training data is generated. (These details would have helped me check the previous point.)
>
> _The ratio coverage classifier is described in Appendix A.2 Setup (page 22) and consists, in our examples, of a simple dense network with 5 times 64 hidden features and ELU activation function. The classifier was trained for just 32 epochs using AdamW optimizer. We updated the description to also say, that the classifier is trained to differentiate between samples generated by the simulator and samples generated by the NPE model. The data is generated identically to the coverage plot algorithms (Algorithm 1 and 2):_
>
> > $\theta_m \sim p(\theta)$ \# prior
>
> > $x_m \sim p(x|\theta_m)$ \# simulator
>
> > $\hat\theta_m \sim q(\hat\theta|x_m)$ \# surrogate model
>
> _The classifier is then trained using a binary cross entropy loss on the pairs $(\theta, x)$ and $(\hat\theta,x)$._
>
> ## Minor
>
> 4. It would be ideal to investigate some more examples. But this is not absolutely necessary in my opinion.
>
> _In order to illustrate that the blind spot problem can also occur for other regularizers from the literature, section A.6 in the appendix was added to the appendix showcasing CalNPE (<font color="green">Falkiewicz et al. (2023)</font>) and BNPE (<font color="green">Delaunoy et al. (2023)</font>)._
>
> 5. Discuss whether the method is only applicable to SBI, or also has applications to other approximate Bayesian methods.
>
> _The focus of this paper is the classical coverage plot, which is used in the SBI literature because it does not require access to the true posterior distribution, which is intractable for most of the relevant applications in SBI. Additionally, the coverage plot offers nice visual interpretability (see Fig. 11). We want to preserve these qualities and make it more powerful by removing the blind spot. Hence, our findings are uniquely targeted towards these cases; however, our fundamental findings hold in general, but they might just not be necessary when other methods are available (e.g., Posterior Predictive Checks)._

---

> ### Author Response · Authors · 2026-01-06
> **Response to Reviewer vyzo**
>
> 6. Use a spelling and grammar checker to fix mistakes. ... there's a sentence fragment that I can't understand.
>
> 7. Many citations would be better in parentheses.
>
> 8. Many equations are unnecessarily split over multiple lines, when they would fit on a single line.
>
> _Citations were moved in parentheses and equations were formatted throughout the manuscript._
>
> 9. Comment on meaning of $\mathcal{B}_\mathcal{Z}$. Is this an arbitrary $\sigma$-algebra? If it's meant to be a Borel $\sigma$-algebra then I think you need to assume something about the topology of $\mathcal{Z}$.
>
> _The $\sigma$-algebra denoted by_ $\mathcal{B}_\mathcal{Z}$ _is considered to be arbitrary (and not related to any topology), when (general) measurable spaces were mentioned. However, for practical purposes, we don't lose anything, if we assume all occurring measurable spaces to be standard Borel spaces (with their Borel $\sigma$-algebra), like the real numbers or real vectorspaces, etc._
>
> 10. On the line before equation (2), I think $\pm$ is a typo of $\pm\inf$.
>
> _The missing symbol was added in Definition A.1._
>
> 11. In Definition 3.4, do you need any extra assumptions to ensure $Q(A|X=x)$ is well defined?
>
> _We presume the reviewer is asking regarding the need for the existence of regular conditional probability distributions $Q(\Theta|X)$. We could have either directly assumed its existence in the premise of the theorem, or that the space for $\Theta$ is a standard Borel space, from which existence follows. However, we only need the existence of $S(t|x)$, which can be defined through conditional expectations, and thus exists on all measurable spaces. If one wanted to show the measurability of $S$, one can solely rely on the fact that $t$ lies in the (extended) real line and $S$ is one-sided continuous in $t$ for $P(X)$-almost-all $x$. Since these are very subtle technical, but standard, arguments, we did not want to bother the reader of this paper with such details._
>
> 12. Page 6 says "we show that the expected conditional coverage gap... is bounded above by the total variation norm". Wasn't this already shown in Equation (19)?
>
> _Yes, we mention it because we use this result to derive the adversarial regularizer in this section. The reworked sentence (page 5) now states: By using that the expected conditional coverage gap for general choice of $g$ is bounded by the total variation norm, we derive an adversarial regularizer for minimizing the coverage gap._
>
> 13. The paragraph before equation (25) mentions that regularity conditions are needed for this result. Can you expand on what these are, and give a citation for where a proof can be found.
>
> _To go from the mathematical equation (Eq. 24) to an optimizable objective we want to replace general measurable functions $f$ by parameterized functions like neural networks. With 'regularity assumptions', we mean sufficiently strong assumptions on $p$ and $q$ such that the corresponding Universal Approximation Theorem for the Neural Networks holds. This requirement can for instance be fulfilled, see Kurt Hornik (1991), for measurable functions in $L^1$-norm w.r.t. some dominating measure $\mu$, e.g. $\mu=(P+Q)/2$. However, in practice, architectures of neural networks are often fixed, and some small gap between theory and practice remains (as in almost all application cases of neural networks in practice). We updated the manuscript to include this reference (page 6)._
>
> 14. Page 6 says "...ratio coverage is based on the observation in Theorem A.4". Can you give more details, as the theorem statement is long and makes several observations.
>
> _The observation is that full distinguishability is achieved for the coverage plot by choosing $g(\theta,x)=log\frac{q(\theta|x)}{p(\theta|x)}$. Theorem A.4 (Theorem A.5 in the reworked manuscript) highlights the relation to the Total variation norm, which is important for two reasons. First, the TV norm is a mathematical sound norm, meaning full discriminative power. Second, it motivates the introduction of the TV regularizer in section 4. We updated the paragraph (page 6) to say that the ratio coverage is motivated by the supremum found in Theorem 2.3 which is $f^\star(z)=$ sgn $g(z)$ with the choice $g(\theta,x):= \log q(z)/p(z)$ (see Remark A.5 and Figure 2)._
>
> 15. On page 7, the paragraph starting "We proofed" seems to repeat some material from Example 3.5.
> If this is intended, perhaps you could add a comment that you're recapping this.
>
> _We clarified this recap in the manuscript (page 7)._
>
> 16. On page 7, is "expected unconditional coverage" a typo?
> Earlier I think you only define "unconditional coverage" and "expected conditional coverage".
>
> _Yes, "expected unconditional" was changed to "unconditional" on page 7._
>
> 17. Appendices A.4 and A.5 should contain some text to explain why you include Figures 10 and 11.
>
> _We added explanations for Figures 10 and 11 to the appendix A.4 and A.5 (page 24 and 25)._

---

> > ### Comment · Reviewer_vyzo · 2026-01-09
> > **Interest to readers**
> >
> > Thanks for the thorough replies to my review, which I will read in detail soon. However, I didn't spot a response to one point - can you comment on this? (Or let me know if the response to another reviewer already covers it.)
> >
> > **Section 4 proposes a novel adversarial regularisation method for NPE, but goes on to show (in Section 6) this has very poor performance. So this method seems of limited interest to readers. Perhaps the authors can add extra justification of why it's important.**

---

> > ### Comment · Reviewer_vyzo · 2026-01-11
> > **Comments on responses**
> >
> > I've checked through the responses, and they satisfactorily answer my substantive points - thanks!. But do see my other recent post about interest to readers.

---

> ### Author Response · Authors · 2026-01-19
> **Second Response to Reviewer vyzo**
>
> We thank the reviewer for their comments and address the point mentioned in "Interest to readers" as follows:
>
> To improve readability, we reworked this section and changed its name to "Classical Coverage Regularization Can Be Deceptive". We clearly state that the proposed adversarial TV regularizer is in line with the expected development in the literature to develop a training objective to optimize for the popular classical coverage metric. But because of its blind spot problem, the regularizer fails to produce good results, although the classical coverage looks better. To show that this result is not limited to our TV regularizer, we included the results from the other regularizers in Section 6.3 in the main text (previously presented in the Appendix), which show the same phenomena. This shows that the problem is not the specific regularizer but the general goal of optimizing for the flawed classical coverage.

---

### Review · Reviewer_LAwX · 2025-12-17

**Summary Of Contributions:**

This paper addresses the assessment of posteriors obtained by simulation-based inference (SBI) methods. While several approaches have been proposed for this purpose, including simulation-based calibration, there remain unresolved issues. To address these, the authors make three main contributions: (i) they provide a general definition of (un)conditional expected coverage, (ii) they propose differentiable objectives that encourage coverage, and (iii) they present an alternative metric of convergence based on density ratios. Finally, the authors demonstrate the effectiveness of their methods through numerical experiments.

**Audience:**

Yes

**Audience Explanation:**

Please see Strength 1.

**Claims And Evidence:**

No

**Claims Explanation:**

Strength 1:
The discriminative property of an evaluation criterion is critically important; therefore, the problem tackled in this paper is highly relevant and significant.


Weakness 1:
I encourage the authors to add a brief review of expected coverage concepts in the previous literature. In the current manuscript, the explanation in Section 3 feels somewhat abrupt. While Example 3.2 shows that the proposed general definition is a natural extension of the classical one, additional contextualization would improve clarity and accessibility.

Weakness 2:
I have strong concerns regarding the practical usefulness of the proposed ratio coverage. As the authors note, ratio coverage requires additional training of a density ratio estimator for evaluation, crucially after observing the data. An important aspect of SBI evaluation is the rapid assessment of posterior quality given observed data. The proposed evaluation metric significantly lacks this property, which makes it appear impractical despite its discriminative power.

**Requested Changes:**

Major points are described in Weakness 1 and Weakness 2.

Minor points:

Sec 3.2: theorem 3.2 => Example 3.2 ?

---

> ### Author Response · Authors · 2026-01-06
> **Response to Reviewer LAwX**
>
> We thank the reviewer for their diligent work, and will address each of the remarks individually:
>
> ### Major
>
> 1. I encourage the authors to add a brief review of expected convergence concepts in the previous literature.
> In the current manuscript, the explanation in Section 3 feels somewhat abrupt.
> While Example 3.2 shows that the proposed general definition is a natural extension of the classical one, additional contextualization would improve clarity and accessibility.
>
> _Presuably, the reviewer refers to the "expected **coverage** concepts" instead of "expected **convergence** concepts".
> In the Introduction, the conventional coverage plot is introduced with respect to the current literature (page 1 and 2):_
> > In lieu of this ideal, fictional algorithm, practitioners typically test the expected conditional coverage of their approximation (<font color="green">Miller et al. (2021); Hermans et al. (2022); Talts et al. (2020); Zhao et al. (2021)</font>). Intuitively, expected conditional coverage tests whether the approximation’s credible region cover the ground truth’s credible region, at all credible levels, averaged over x ∼ p(x). Although this sounds promising as a criterion to determine whether two distributions agree, there are issues with expected conditional coverage. One known limitation is that the prior p(θ) has exact expected conditional coverage at all credibility levels <font color="green">Delaunoy et al. (2023); Lemos et al. (2023)</font>, i.e. this method cannot distinguish between p(θ) and p(θ | x). Despite this limitation, many works have proposed algorithms aimed at improving the expected conditional coverage of learned posteriors through regularization (<font color="green">Delaunoy et al. (2022; 2023); Dey et al. (2022); Falkiewicz et al. (2023)</font>). We will propose another one.
>
> _Then, in Section 3.2 we give the formal definition using the framework of the general coverage, where the conventional definition corresponds to the _expected conditional coverage_ with the classical choice of $g=q$. Our definition of the classical coverage thereby follows the choices in <font color="green">Hermans et al. (2022)</font>._
>
> 2. I have strong concerns regarding the practical usefulness of the proposed ratio convergence.
> As the authors note, ratio convergence requires additional training of a density ratio estimator for evaluation, crucially after observing the data.
> An important aspect of SBI evaluation is the rapid assessment of posterior quality given observed data.
> The proposed evaluation metric significantly lacks this property, which makes it appear impractical despite its discriminative power.
>
>
> _Presuably, the reviewer refers to the "ratio **coverage**" instead of "ratio **convergence**".
> We thank the reviewer for this question, as it highlights another result of our paper.
> Concerning the classifier, our method and the theory hold for general discriminating function, i.e., even if the ratio is not fitted perfectly, the ratio coverage has a high discriminating power. In all our test cases, a simple classifier and a short training were sufficient to achieve these results (see Appendix A.2 Setup), and other factors like sampling time (which applies to classical and ratio coverage) took the majority of the evaluation time. The bottleneck, in terms of computational cost and performance, for simulation-based inference in our experiments was not the classifier, but the training of the NPE model itself. Other methods in the literature, like $\ell$-C2ST and DiscCali, have equal or even higher computational cost when training their classifiers.
> However, a major improvement in performance is using the unconditional coverage as an alternative to the expected conditional coverage (see Algorithm 2 and 1), which is the standard for the classical coverage in the literature. Our comparison shows that the unconditional coverage gives similarly good plots and reduces the computational cost of the evaluation. These results are discussed in section 5.3 Unconditional Coverage and A.4 in the Appendix._
>
> ### Minor
>
> 3. Sec 3.2: theorem 3.2 $\Rightarrow$ Example 3.2 ?
>
> _We fixed some incorect references in the manuscript, where the reference was named "Theorem" while the linke refered to an "Example"._
>
> We hope that the reviewer finds our explanations satisfactory, and we want to thank him again for contributing to our publication.

---

> ### Comment · Reviewer_LAwX · 2026-01-13
>
> Thank you for your detailed response, and sorry for the confusion—I meant “coverage” instead of “convergence.”
>
> Regarding the reply to Weakness 1, thank you for the clarification. However, even after this explanation, I still agree with Reviewer ojnF and find the argument difficult to follow, mainly because the explanation remains too mathematical. Researchers interested in SBI come from a wide range of fields, so I believe the exposition should be further clarified to be accessible to a broader audience.
>
> For example, in the current presentation, the authors first introduce the general definition (Definition 3.1) and then show that it includes the classical case (Example 3.2). While this is mathematically sound, the explanation might be more readable if the intuition behind Figure 1 were first described in mathematical terms, and Definition 3.1 were then derived as a generalization of that intuition, or if authors add more explanations why g = q yields the Fig. 1 to Example 3.2.  (This is what I intended to suggest in my original review).
>
> Regarding Weakness 2,
> I understand that training an additional classifier is essentially the same as performing a C2ST. While C2ST can be applied when the true distribution is known, this corresponds to a different setting from the one addressed in this paper. Nevertheless, I understand that the proposed criteria still have merit, even though they require the cost of training an additional classifier, similar to C2ST. Thanks.

---

> ### Author Response · Authors · 2026-01-19
> **Second Response to Reviewer LAwX**
>
> We thank the reviewer for the kind words and continuous work on the manuscript. We addressed the remaining point as follows:
>
> To simplify the explanations, we replaced Figure 1 to explain in detail how the coverage plot is constructed and how the blind spot problem develops. Figure 1 explicitly considers three cases: (1) the model and true distribution agree, (2) both distributions are different and (3) the blind spot problem. It also shows and introduces graphically some of the terms later used in the mathematical derivation, e.g., super-level sets. In Section 3, we moved the Remark "Estimating the expected conditional coverage plot" to the Appendix and instead moved the Algorithms 1 and 2 into the main text. To highlight, that the bad performance of the regularized models is not specific to our TV regularizer, we included the results for regularizers from the literature into Section 6.3 (previously found in the Appendix). This shows that optimizing for the flawed classical coverage is the cause of the bad performance.

---

### Review · Reviewer_ojnF · 2025-12-19

**Summary Of Contributions:**

**Summary:** This paper addresses the challenge of evaluating posterior approximations in SBI, as the ground-truth posterior is not available. In this context, classical metrics based on two-sample testing (MMD, C2ST, etc.) are not applicable. In the SBI literature, one solution is typically to perform expected coverage tests (ECT, a.k.a. simulation-based calibration - SBC), but it is common knowledge that these tests do not present a necessary condition to guarantee the correctness of the posterior approximation.
The authors mention the well known "prior" blind-spot (the prior will always pass the ECT) and additionally claim to have found a second blind spot, namely that the ECT can even pass for approximations that put mass in unwanted regions-where the ground-truth posterior has none. Still, many works have focused on regularizing SBI methods to enforce good expected coverage. In light of this trend, the authors propose an alternative regularization method with the same purpose.
However, the main contribution of this paper is a new way to assess coverage that addresses the blind-spots of the classical ECT. The authors introduce the notion of "ratio coverage" and claim that it is "discriminative", i.e. a sufficient condition to guarantee the correctness of the posterior approximation. However, it comes at the cost of estimating the ratio between the approximated and true posteriors (e.g. by training a binary classifier), which is the main limitation of the proposal.

**Key strengths:**
- The authors address an important topic in SBI
- The problem statement is well formulated in the introduction
- The authors provide rigorous mathematical definitions of unconditional and expected conditional coverage
- The main claims are supported with theoretical and empirical evidence through theorems and numerical experiments

**Key weaknesses:**
- The paper is not easy to read (especially the theoretical sections)
- Some major clarifications are needed to understand the main contribution - ratio coverage: how it is indeed discriminative and how it works in practice
- The experiment section lacks comparisons to existing work
- Some key references are missing (local coverage tests)

**Additional Comments:**

I understand that this review is long and that not all the questions can be answered in detail. My main point is: the proposal is interesting. However, it currently requires significant clarification, a more complete discussion of related work (in particular local coverage plots), and a major effort to improve clarity and accessibility before it can be considered a strong and impactful contribution.

**Audience:**

Yes

**Audience Explanation:**

I think people working in the field of SBI would find this work interesting, due to its new perspective on the very popular coverage plots and their interesting and enlightening empirical findings about how regularization can hurt accuracy.

**Claims And Evidence:**

No

**Claims Explanation:**

The claims are supported by interesting empirical evidence: the experiments show the "discriminative" nature of the proposed ratio-based coverage method, as opposed to the classical coverage, which fails to detect bad posterior approximations. They also show how their regularization improves the expected coverage, but can hurt the accuracy of the learned posterior. However some clarifications about the experimental setup and more exhaustive experiments about the sensitivity of the method would make it more convincing.

The claims are also supported by accurate theoretical evidence. However, the lack of clarity makes them less convincing.

I have some important questions that i'd like to be addressed!

**Requested Changes:**

### 1. Major clarifications about the proposal
- When introducing KL, KS and TV, it is not clear why and where this is going to show up / to be relevant  in the rest of the article. I suggested making that more explicit. To be more specific: it would be important to clarify / explicit the relationship between the ratio coverage and  the other divergences. The authors mention it at several places, but never in a concise and clear way. Also, the KS only shows up in the appendix. I would have liked to have a clear paragraph that gives the reader a reference point to hold on to and that guides him through the ideas of the paper.
- I like the example 3.2, it is crucial to understand what's happening. Here, the authors mention the blind-spot as a consequence to choosing $g=q$. Could the authors explain further why that happens?
- In section 5, the authors mention that in practice, one has to train a classifier to estimate the ratio. They also state that estimating the unconditional ratio coverage plot is simpler (in terms of derivations and computational cost) than the expected conditional one. I understand what they mean, but it would be good for readers to have more explanations on how they do that and what in what setting unconditional coverage is useful / interesting to look at. If I'm not mistaken, the unconditional version is not used in the experiments. Why is that?
- Again in Section 5, the authors make a comment about the NPE objective being an upper bound to the adversarial ratio regularizer introduced in Section 4, which can thus be "dropped from the NPE training objective". Does this mean that the regularizer has no effect in the NPE case? Or even that it harms optimization in terms of improving the accuracy of the posterior approximation? I believe this is shown in the experiments in Section 6.1. If that's the case it would be good to refer to those. As is, it is unclear why they make that comment.

### 2. Minor clarifications about the proposal
- In remark 3.6 the authors relate the coverage gap to the total variance. First, I understand that this is important for section 4., but could also be put in the appendix. Second, aren't equations (19) and (22) the same thing? I thought this was a direct consequence of Theorem 2.4 in Section 2. Or am I missing something?
- In section 4, the authors mention that $f$ can be restricted to *sufficiently flexible neural network with tanh outputs*. What does *sufficiently flexible* mean, and why is that important for us? Why tanh outputs? I don't know how relevant these informations are for the main paper, but if the authors are going to give the reader this information, they should also say why.
- In Definition 3.1, the authors mention that $g$ is a *discriminating function*, without specifying what that is supposed to mean or how relevant that is. I suggest clarifying that. Same goes for the mention of the super-level set, even if some intuition is given in the example below.
- Definition 3.4 assumes $P$ and $Q$ to have the same marginals. I know it is a classical assumption in SBI, but in might be worth mentioning why that is (in a footnote or so).

### 3. More intuition and accessibility for theoretical results
- In Section 3, is the main reference for definitions (with choices of notations and names for used quantities) always Hermans et al.? Or are the authors taking liberties and / or notations from other references? Given the title of the section "*generalizing* coverage plots", I would expect they are taking liberties, but it would be good to know what their additions are w.r.t. what already exists in the literature.
- I find Section 3.1 very math heavy and not easy too understand for the general SBI community.  I believe in the main paper, things should be explained in a less heavy, more intuitive and more practical way. The math can be put in the appendix for people who are looking for a deeper understanding.
For instance, I like the remark 3.3. about how the plots are computed with samples, but it is still not very easy to digest and not obvious how things are done in practice. The authors refer to Algorithm 2 for more information, which I find very helpful, especially to better understand what is plotted (and against what) in the end. Why not put the algorithm here? Or at least explain how the plotting in algorithm compares to what is explained in remark 3.3 (instead of computing the super-level sets twice, for $P$ and $Q$, algorithm computes it only once, for $Q$, as samples from $P$ are used to define the thresholds, meaning that the number of samples from $P$ above those thresholds will be in the range $[1,N]$).
- Again, in definition 3.4, I find the notations not trivial and difficult to understand. I believe it could be formulated much easier or authors could give more intuition (e.g. $S$ is related to the c.d.f. !). The authors clarify things in Example 3.5 by saying that expected conditional coverage, as the name indicates, is just taking the expectation of the conditional coverage over $x$. Why not reflect that in the math by using (conditional) expectations? (i believe this what Zhao et al. did in their definition of Local Coverage Plots for example)
- Theorem 5.1 - in particular Equations (19) and (20) - is the main result and main contribution of this paper from my point of view. It would be valuable to at least have a sketch of the proof here in the main part of the paper, or some intuition on why that is true.
- In the inequalities in Theorem 5.1. the authors use a new notation ($g$ _# $P$) without introducing it. This notation is only introduced in the Appendix, but should be introduced when they first use it in the main paper. Also, it is not clear to me what it means. This made understanding the proof very difficult. On that note, I second my comment from before about wanting an another version of the proof that is easier to understand / more intuitive.

### 4. Experiments
- Have the authors compared their method (TV-regularization) to other regularization techniques, like the one from Hermans et al. (or the other techniques mentioned in the introduction)?
- The problem pointed out in 6.1 about regularized models not improving the posterior approximation for larger training sets is very interesting! I really like how visible this is in the diagonal subplots from Fig 4. where the regularization clearly results in a "mode collapse" (for 1,2 and 3 modes corresponding to diagonal subplots 2,3 and 4). I suggesting highlighting that by referring to the figure!
- Another question about this blind-spot detected in 6.1: it concerns the "prior" blind-spot that is very well known in SBI. However the authors claimed to have found another blind-spot, which they illustrate in Figure 1. Could that also happen with their regularization technique? Do the authors have any idea what example one could use to show that empirically?
- In 6.2, how many samples are used to train the classifier? How sensitive is the method to the sample size? Some additional experiments, similar to what has been done in Linhart et al. for L-C2ST (or just referring to their work), would be important to convince the readers about the effectiveness of the proposed method.
- It would be interesting to compare the unconditional and the expected conditional ratio coverage plots. Why did the authors choose not to do that?

### 5. Related work
- The paper seems to be missing an important part of the literature on _local_ coverage tests (see for example Zhao et al., 2021: Diagnostics for Conditional Density Models and Bayesian Inference Algorithms). It would be important for the authors to discuss how their approach relates to these local coverage tests, both conceptually and practically, and to clarify what is genuinely new compared to that line of work. Relevant references seem to be missing.
- The authors highlight the related work form Linhart et al., L-C2ST, which they explain is the same thing but defined differently (not in terms of coverage). Linhart et al. talk about "consistency", which they define as $p=q$  a.e. Isn't that closely related to coverage? From what i understand from this paper, this is what perfect coverage is aiming for and what they claim providing with the ratio coverage.  In fact, couldn't the ratio function  introduced in this work be used for exactly the same purpose as the L-C2ST classifier? And reversely, couldn't the L-C2ST classifier be used as the $g$ function in the ratio plot? In other words, isn't it exactly the same thing, as the discriminator function of the classifier implicitly defines the ratio between p and q? Finally, the authors say that ratio coverage implies passing L-C2ST for all $x_0$. Is that so? Could the authors clarify? Only for the $x_0$s used in the ratio coverage plot, no? It seems obvious to me that the reverse is true aswell. If their statement is true, then that confirms that both methods are equivalent, just introduced in a different context and for a different purpose.
- In section 5, the authors conclude with a comment on ratio coverage seen from a hypothesis testing point of view, which appears to be very relevant as they claim learning the most powerful test statistic. This paragraph lacks references to strengthen their argument. Also I suggest highlighting / developing this comparison. For example it is unclear to me what the authors mean by "our test should determine which of them is more likely". How does this relate to the coverage plot? I can think of an interpretation, but it would be nice to give the reader more intuition.

### 6. Typos and referencing issues
- In 3.2, the authors refer to a "Theorem 3.2". Am I missing something or does this theorem not exist? Also, I believe there is a typo or something missing in the first paragraph when the authors write "(theorem 3.2), which is used in ...", this is not a real sentence.
- In Section 5, the authors mention a Theorem A.4 that doesn't exist. maybe they meant Theorem 2.4 or Remark A.4? Also if they mention it, they should say why / what the observation is that they are referring to.
-  Figure 2 is never mentioned!
- I guess the "s" is missing in the section 6 title "Experiment"?

---

> ### Author Response · Authors · 2026-01-06
> **Response to Reviewer  ojnF**
>
> We thank the reviewer his extensive analysis of our work and will respond to all remarks individually:
>
> ### Major
>
> 1. When introducing KL, KS and TV, it is not clear why and where this is going to show up / to be relevant in the rest of the article. I suggested making that more explicit. To be more specific: it would be important to clarify / explicit the relationship between the ratio coverage and the other divergences. The authors mention it at several places, but never in a concise and clear way. Also, the KS only shows up in the appendix. I would have liked to have a clear paragraph that gives the reader a reference point to hold on to and that guides him through the ideas of the paper.
>
> _The KL divergence is commonly used as a training objective for NPE models and is also the foundation for our proposed regularizer (see Section 4).
> The TV distance is related to an optimality condition for the classical coverage and subsequently is used for our "Adversarial Total Variation Norm Regularization" (see Section 4). We moved the KS divergence to the appendix and added brief clarifications for the other definitions to the manuscript (page 14 and 3)._
>
> 2. I like the example 3.2, it is crucial to understand what's happening. Here, the authors mention the blind-spot as a consequence to choosing $g=q$. Could the authors explain further why that happens?
>
> _The intuitive explanation why this choice allows for a blind spot is that the function $g$ is independent of $p$. As Figure 1 in the manuscript illustrates, the classical coverage cannot detect any probability mass of $p$ in a region where $q$ has a small probability._
>
> 3. In section 5, the authors mention that in practice, one has to train a classifier to estimate the ratio. They also state that estimating the unconditional ratio coverage plot is simpler (in terms of derivations and computational cost) than the expected conditional one. I understand what they mean, but it would be good for readers to have more explanations on how they do that and what in what setting unconditional coverage is useful / interesting to look at. If I'm not mistaken, the unconditional version is not used in the experiments. Why is that?
>
> _Throughout most of our experiments, we contrast the classical and ratio coverage using Algorithm 1, the expected conditional coverage plot. Only in the last experiment, Section 6.3 Unconditional Coverage, we instead compare Algorithm 1 with Algorithm 2, the Generalized unconditional coverage plot, and conclude that Algorithm 2 reduces the computational cost while producing nearly identical coverage plots. We make sure to specify the used technique in the caption of each figure. In the Appendix A.4, we investigate the difference between both methods further, and test how the number test samples ($N$ and $M$) influences the reliability of the plot. In the Conclusion we summarize our findings that Algorithm 2 should be considered a computational less costly alternative to the expected conditional coverage that is used in the literature._
>
> 4. Again in Section 5, the authors make a comment about the NPE objective being an upper bound to the adversarial ratio regularizer introduced in Section 4, which can thus be "dropped from the NPE training objective". Does this mean that the regularizer has no effect in the NPE case? Or even that it harms optimization in terms of improving the accuracy of the posterior approximation? I believe this is shown in the experiments in Section 6.1. If that's the case it would be good to refer to those. As is, it is unclear why they make that comment.
>
> _The proposed regularizer from Section 4 (Equation (24)) is based on the classical coverage definition $g(\theta, x) := \log q(\theta|x)$. Deriving an corresponding regularizer for the ratio coverage $g(\theta, x) := \log q(\theta, x)/p(\theta, x)$ is however not usefull, since we show in Section 5 (Equation (29)) that the KL divergence (our base loss function) is already a tight bound in this case and can be "dropped from the NPE training objective". More explanations and references were added to the manuscript (page 7 and 8)._
>
>
> ### Minor
>
> 1. In remark 3.6 the authors relate the coverage gap to the total variance. First, I understand that this is important for section 4., but could also be put in the appendix. Second, aren't equations (19) and (22) the same thing? I thought this was a direct consequence of Theorem 2.4 in Section 2. Or am I missing something?
>
> _The remark 3.6 is split has two parts (equation (19) and (22)) to first show the general idea for $P(\Theta, X)$ and then specify the bound for the conditional probability case, e.g. $P(\Theta | X)$, where we assume that $P(X) = Q(X)$. This specification is necessary because the coverage is conventionally defined in the conditional case and not for the joint probability._

---

> ### Author Response · Authors · 2026-01-06
> **Response to Reviewer ojnF**
>
> 2. In section 4, the authors mention that $f$ can be restricted to _sufficiently flexible neural network with tanh outputs_. What does _sufficiently flexible_ mean, and why is that important for us? Why tanh outputs? I don't know how relevant these informations are for the main paper, but if the authors are going to give the reader this information, they should also say why.
>
> _To go from the mathematical equation (Eq. 24) to an optimizable objective we want to replace general measurable functions $f$ by parameterized functions like neural networks. This also relates to the flexibility of the model and the amount of data available for training. In practice, the training of the model is always limited, either by the available data, the flexibility of the model or the sensitivity of the objective. If data and/or flexibility are limiting the training, then in the SBI setting more data can be generated using the simulator, which directly corresponds to an increase in computing cost.
> With 'regularity assumptions', we mean sufficiently strong assumptions on $p$ and $q$ such that the corresponding Universal Approximation Theorem for the Neural Networks holds. This requirement can for instance be fulfilled, see Hornik 1991, for measurable functions in $L^1$-norm w.r.t. some dominating measure $\mu$, e.g. $\mu=(P+Q)/2$. However, in practice architectures of neural networks are often fixed, and some small gap between theory and practice remains (as in almost all application cases of neural networks in practice). The citation was added to page 6._
>
>
> 3. In Definition 3.1, the authors mention that $g$ is a _discriminating function_, without specifying what that is supposed to mean or how relevant that is. I suggest clarifying that. Same goes for the mention of the super-level set, even if some intuition is given in the example below.
>
> _In the manuscript, we introduce the concept of the coverage plot with an intuitive explanation in the introduction (page 1 and 2). We explain that choosing the credibility regions averaged over $x \sim p(x)$ has a known limitation as it cannot distinguish between $p(\theta)$ and $p(\theta | x)$. To address this issue, we propose a generalized version of coverage that ( a ) discriminates between distributions $q$ and $p$, i.e., it only returns the typical diagonal line when $q=p$, ( b ) does not require samples from the ground truth posterior, and ( c ) does not suffer from either blind spot mentioned above. However, it comes at the cost of estimating the ratio $\frac{q(\theta \mid x) p(x)}{p(\theta, x)} = \frac{q(\theta \mid x)}{p(\theta \mid x)}$. In Section 3 and Definition 3.1 we formalize this idea._
>
> 4. Definition 3.4 assumes $P$ and $Q$ to have the same marginals. I know it is a classical assumption in SBI, but in might be worth mentioning why that is (in a footnote or so).
>
> _In the Introduction we describe the basic ingredients of SBI with the surrogate model $q$, which is an approximation of the true distribution $p$. We added an explanation and reference in the manuscript (page 4)._
>
> 5. In Section 3, is the main reference for definitions (with choices of notations and names for used quantities) always Hermans et al.? Or are the authors taking liberties and / or notations from other references? Given the title of the section "generalizing coverage plots", I would expect they are taking liberties, but it would be good to know what their additions are w.r.t. what already exists in the literature.
>
> _We start in the Introduction with the definitions and names from the current literature and motivate a new definition to fix the known problems of the coverage plot. In Section 3, we give a more general definition which leads to the classical definition as explained in Example 3.2, but also enables our proposed ratio coverage (see Section 5)._
>
> 6. I find Section 3.1 very math heavy and not easy too understand for the general SBI community. I believe in the main paper, things should be explained in a less heavy, more intuitive and more practical way. The math can be put in the appendix for people who are looking for a deeper understanding. For instance, I like the remark 3.3. about how the plots are computed with samples, but it is still not very easy to digest and not obvious how things are done in practice. The authors refer to Algorithm 2 for more information, which I find very helpful, especially to better understand what is plotted (and against what) in the end. Why not put the algorithm here? Or at least explain how the plotting in algorithm compares to what is explained in remark 3.3 (instead of computing the super-level sets twice, for $P$ and $Q$, algorithm computes it only once, for $Q$, as samples from $P$ are used to define the thresholds, meaning that the number of samples from $P$ above those thresholds will be in the range $[1,N]$).

---

> ### Author Response · Authors · 2026-01-06
> **Response to Reviewer ojnF**
>
> 7. Again, in definition 3.4, I find the notations not trivial and difficult to understand. I believe it could be formulated much easier or authors could give more intuition (e.g. $S$ is related to the c.d.f. !). The authors clarify things in Example 3.5 by saying that expected conditional coverage, as the name indicates, is just taking the expectation of the conditional coverage over $x$. Why not reflect that in the math by using (conditional) expectations? (i believe this what Zhao et al. did in their definition of Local Coverage Plots for example)
>
> _Also giving practical examples and connection formulas to practice is an important contribution of this manuscript is to clearly formulate mathematically what is currently done in practice, e.g., the classical coverage, and why it works. Based on this, we are able to explain the blind spot problem, i.e., why the coverage cannot distinguish $p(\theta)$ from $p(\theta | x )$. Hence, throughout the manuscript, we are mathematically explicit about what we do, in contrast to other papers on the topic._
>
> 8. Theorem 5.1 - in particular Equations (19) and (20) - is the main result and main contribution of this paper from my point of view. It would be valuable to at least have a sketch of the proof here in the main part of the paper, or some intuition on why that is true.
>
> _The choice of $g$ for the ratio coverage is motivated by Theorem 3.2 and Equation (19) in the main manuscript, which finds the log ratio of $q$ and $p$ as the supremum in the TV norm, which is a bound on the coverage gap. We clarified this in the manuscript (page 6) by stating that the ratio coverage is motivated by the supremum found in Theorem 2.3 which is $f^\star(z)=$ sgn $g(z)$ with the choice $g(\theta,x):= \log q(z)/p(z)$ (see Remark A.5 and
> Figure 2)._
>
> 9. In the inequalities in Theorem 5.1. the authors use a new notation ($g_\#~P$) without introducing it. This notation is only introduced in the Appendix, but should be introduced when they first use it in the main paper. Also, it is not clear to me what it means. This made understanding the proof very difficult. On that note, I second my comment from before about wanting an another version of the proof that is easier to understand / more intuitive.
>
> _We changed the manuscript to define the notation where it first appears in Remark 3.6 (page 5)._
>
> 10. Have the authors compared their method (TV-regularization) to other regularization techniques, like the one from Hermans et al. (or the other techniques mentioned in the introduction)?
>
> _To show that this blind spot problem is not specific to our TV regularizer, we compared related methods CalNPE (Falkiewicz et al. (2023)) and BNPE (Delaunoy et al. (2023)) with our TV regularized models, trained comparable to the experiments in the main text. All regularization methods confirm our previous results. We see that for all regularized models, the classical coverage is improved (near perfect), but the posterior distribution is clearly different from the true posterior. In contrast to the classical coverage, the ratio coverage is able to detect this correctly and identifies the unregularized model as the best result, which aligns with our observation from the comparison to the true posterior. Section A.6 and Figure 12 were added to the appendix (page 25) containing this result._
>
> 11. The problem pointed out in 6.1 about regularized models not improving the posterior approximation for larger training sets is very interesting! I really like how visible this is in the diagonal subplots from Fig 4. where the regularization clearly results in a "mode collapse" (for 1,2 and 3 modes corresponding to diagonal subplots 2,3 and 4). I suggesting highlighting that by referring to the figure!
>
> _We changed the manuscript (page 8) to state the regularized models show no improvement for larger training sizes and, in general, have less conformance with the true posterior. This is especially visible for the diagonal plots in Figure 4. The models without regularization perform better than the models with regularization, which confirms that the regularizer harms the optimization (see Section 5)._
>
> 12. Another question about this blind-spot detected in 6.1: it concerns the "prior" blind-spot that is very well known in SBI. However the authors claimed to have found another blind-spot, which they illustrate in Figure 1. Could that also happen with their regularization technique? Do the authors have any idea what example one could use to show that empirically?
>
> _In the literature, it is well known that the prior can give ideal classical coverage plots, which we also observe in the Experiments (Figure 4). On the bottom of page 8, we discuss this fact and also show that the ratio coverage **is able** to resolve this problem. Specifically, Figure 7 in the Appendix illustrates how this problem can be understood as a special case of the blind spot problem that we discuss in the main paper._

---

> ### Author Response · Authors · 2026-01-06
> **Response to Reviewer ojnF**
>
> 13. In 6.2, how many samples are used to train the classifier? How sensitive is the method to the sample size? Some additional experiments ... to convince the readers about the effectiveness of the proposed method.
>
> _Concerning the classifier, our method and the theory hold for general discriminating function, i.e., even if the ratio is not fitted perfectly, the ratio coverage has a high discriminating power. In all our test cases, a simple classifier and a short training were sufficient to achieve these results (see Appendix A.2 Setup), and other factors like sampling time took the majority of the evaluation time. The bottleneck, in terms of computational cost and performance, in our experiments was not the classifier, but the training of the NPE model itself._
>
> 14. It would be interesting to compare the unconditional and the expected conditional ratio coverage plots. Why did the authors choose not to do that?
>
> _As discussed above, this comparison can be found in Section 6.3 "Unconditional Coverage"._
>
> 15. The paper seems to be missing an important part of the literature on local coverage tests (see for example Zhao et al., 2021: Diagnostics for Conditional Density Models and Bayesian Inference Algorithms). It would be important for the authors to discuss how their approach relates to these local coverage tests, both conceptually and practically, and to clarify what is genuinely new compared to that line of work. Relevant references seem to be missing.
>
> _In the introduction, we discuss other diagnostic tools, also with a reference to this work. Specifically in the related work section we discuss local coverage tests like $\ell$-C2ST, which "is more sensible, precise, and computationally efficient than its concurrent method, local-HPD." (from 5 Discussion in Linhart et al. (2023)) where local-HPD referes to the afore mentioned Zhao et al. (2021)._
>
> 16. The authors highlight the related work form Linhart et al., L-C2ST, which they explain is the same thing but defined differently (not in terms of coverage). Linhart et al. talk about "consistency", which they define as $p=q$ a.e. Isn't that closely related to coverage? From what i understand from this paper, this is what perfect coverage is aiming for and what they claim providing with the ratio coverage. In fact, couldn't the ratio function introduced in this work be used for exactly the same purpose as the L-C2ST classifier? And reversely, couldn't the L-C2ST classifier be used as the $g$ function in the ratio plot? In other words, isn't it exactly the same thing, as the discriminator function of the classifier implicitly defines the ratio between p and q? Finally, the authors say that ratio coverage implies passing L-C2ST for all $x_0$. Is that so? Could the authors clarify? Only for the $x_0$s used in the ratio coverage plot, no? It seems obvious to me that the reverse is true aswell. If their statement is true, then that confirms that both methods are equivalent, just introduced in a different context and for a different purpose.
>
> _The generalized framework that is introduced in our manuscript is purposefully general. Its formulation allows for general discriminating functions $g$ such that we can compare different cases in one framework (e.g., classical coverage, ratio coverage, etc). So, every other function $g$, however trained, can be used in this framework as well (with probably different properties). The L-C2ST seem to train the same classifier as we do, but only report one number, while we plot the whole graph and connect it to the whole coverage plot framework._
>
> 17. In section 5, the authors conclude with a comment on ratio coverage seen from a hypothesis testing point of view, which appears to be very relevant as they claim learning the most powerful test statistic. This paragraph lacks references to strengthen their argument. Also I suggest highlighting / developing this comparison. For example it is unclear to me what the authors mean by "our test should determine which of them is more likely". How does this relate to the coverage plot? I can think of an interpretation, but it would be nice to give the reader more intuition.
>
> _We reworked the manuscript (page 7) to state clearly that in a hypothesis testing framework, we consider a sample $z$ drawn from either $P$ or $Q$, and our test tries to determine from which distribution it originates most likely. In a scenario where the null hypothesis is $P$ and the alternative is $Q$, our unconditional ratio coverage plot directly plots the typeIerror versus the typeIIerror._
>
> 18. In 3.2, the authors refer to a "Theorem 3.2" ... this is not a real sentence.
>
> 19. In Section 5, the authors mention a Theorem A.4 that doesn't exist ....
>
> _We corrected the naming and removed the sentence fragment._
>
> 20. Figure 2 is never mentioned!
>
> _References were added in section 5 (page 6)._
>
> 21. I guess the "s" is missing in the section 6 title "Experiment"?
>
> _The section title was changed._

---

> > ### Comment · Reviewer_ojnF · 2026-01-09
> > **Response**
> >
> > I would like to thank the authors for their detailed response and the provided clarifications / additions to the original version of the paper.
> >
> > While this goes into the right direction, I do think more changes are required to make this submission acceptable for publication.
> >
> > - In particular, my main concern about **less heavy mathematical parts with simpler notations** and **more intuition on the practical use of the method** are crucial for the clarity of the paper and would make it more convincing (questions 6 and 7 for example were not properly addressed).
> > - Furthermore, it seems that some less relevant parts (e.g. the flexibility of the neural network functions) could be removed or moved to the appendix to **make space for more important things** like a sketch of the proof for Theorem 5.1 (Equation 26) - which is at the heart of the contribution of this work - as I suggested in question 8.
> > - Finally, it would be good to have **access to the code in the spirit of reproducibility and open source**.

---

> > > ### Comment · Reviewer_ojnF · 2026-01-09
> > > **Response (detailed)**
> > >
> > > To be more precise, here are my comments for remaining typos, missing clarity and suggested changes:
> > >
> > > 1. The part about the divergences is way better, thanks for adding the clarifications! I think it's good that you moved the KS to the appendix. Careful though, it still appears in Remark 3.6.
> > >
> > > 2. Example 3.2: I suggest replacing "it cannot generally distinguish between P and Q" with the more explicit comment from their response: "the classical coverage cannot detect any probability mass of in a region where has a small probability."
> > >
> > > 3. About the adversarial ratio regularizer in Section 5: do I understand correctly that it makes no sense for NPE? So the conclusion is the ratio regularizer is useless as opposed to the classical regularizer introduced in section 4? Also, going in the same direction as Reviewer vyzo: why introduce the regularizer if it shows the same (bad) results as the already existing regularizers (thanks for adding the experiments in Figure 12!!). I was also wondering if the regularizer necessary yields a distribution resembling the prior or if it could be something else (question 12).
> > >
> > > 4. Notations: The "push-forward" function notation introduced in Remark 3.6 is still very confusing to me. I would really simplify that. In general, as mentioned in my review, I would clarify whether notations are taken from the literature (with references!) or not.
> > >
> > > 5. Clarifications: In my opinion it is still important to define / remind the reader what a discriminative function is in Definition 3.1. A footnote would be enough. Also, the footnote saying that the surrogate model is such that P(x)=Q(X) refers to section 1, but you never explain why we actually have P(X)=Q(X) (or why it is okay to assume that). I suggest adding a sentence in the footnote clarifying that.
> > >
> > > 6. Practical use of the method: You never explicitly mention how many samples are needed to train (and evaluate) the classifier. I second my question 13, it is important to include all experimental details and justify choices. At the very least you should refer to other work, e.g. the l-c2st paper which by the way shows that the sample size for the classifier is not trivial at all and can severely influence the discriminative power of the method.
> > > In general, I feel like you could make space to better explain the practical workflow of your method. I do understand that the theoretical framework is part of your contribution, but you are claiming to provide a diagnostic tool that is practical to use, so there should not be any open questions left after reading the paper. As I mentioned in my review, this would also clarify the theory and make the whole thing more convincing (e.g. putting the algorithms in the main paper to illustrate the definitions of coverage and connect both theory and practice).
> > >
> > > 7. Missing literature and related work: I still think a more explicit connection to local coverage plots (Zhao et al.) is missing. Especially as you mention that L-C2ST "does not specifically address coverage". You could for example just add the citation about L-C2ST you wrote in your response: "is more sensible, precise, and computationally efficient than its concurrent method, local-HPD." As the ratio coverage is more or less equivalent to L-C2ST, that strengthens your claims and makes your proposal more convincing. Also, I do agreee with your response to question 16, it is nice to have a connection to coverage and a interpretable plot in addition to the L-C2ST. However, the last sentence in the "Related work" paragraph suggests that the ratio coverage is stronger ("An estimator with ratio coverage would pass L-C2ST for all x_o"), but it actually goes both ways as both methods are equivalent. Also I do not necessarily agree, it would only pass the L-C2ST for the x_o used for the ratio coverage plot, for any other x_o there is no guarantee. This is where L-C2ST is powerful, as it can test for individual (new) observations x_o.
> > > Finally, you mention the related work from Dirmeier et al. at the same place as L-C2ST from Linhart et al. This is confusing: you don't explain what they do, so one might think that they are also part of the L-C2ST work. Please clarify that or remove this reference.

---

> ### Author Response · Authors · 2026-01-19
> **Second Response to Reviewer ojnF**
>
> We want to thank the reviewer for their constant work and commitment to improving the manuscript.
>
> We addressed the recommendations as follows:
>
> 1. In particular, my main concern about less heavy mathematical parts with simpler notations and more intuition on the practical use of the method are crucial for the clarity of the paper and would make it more convincing (questions 6 and 7 for example were not properly addressed).
>
> 2. Furthermore, it seems that some less relevant parts (e.g. the flexibility of the neural network functions) could be removed or moved to the appendix to make space for more important things like a sketch of the proof for Theorem 5.1 (Equation 26) - which is at the heart of the contribution of this work - as I suggested in question 8.
>
>  _In order to follow the reviewers' recommendation to make the manuscript more intuitive and easier to understand, we replaced Figure 1 to explain more descriptively how the equation can be understood and what different terms mean, e.g. super-level sets. We moved the Remark about Estimating the expected conditional coverage plot to the appendix and instead moved Algorithm 1 and 2 into the main text._
>
> 3. Finally, it would be good to have access to the code in the spirit of reproducibility and open source.
>
> _All code to reproduce the results presented in this paper is available: https://anonymous.4open.science/r/ratio_coverage-E777/ . The link is added to the Section A.2 Setup of he paper where also the implementation details are discussed._
>
> 4. The part about the divergences is way better, thanks for adding the clarifications! I think it's good that you moved the KS to the appendix. Careful though, it still appears in Remark 3.6.
>
> _Remark 3.6 was adjusted to refer to the definition of KS:
> with ..., KS the Kolmogorov–Smirnov divergence (Definition A.1) and ..._
>
> 5. Example 3.2: I suggest replacing "it cannot generally distinguish between P and Q" with the more explicit comment from their response: "the classical coverage cannot detect any probability mass of in a region where has a small probability."
>
> _In order to keep the formal meaning of the sentence  intact and incorporate the suggested phrasing, we changed the sentence to: "This choice for $g(z)$ comes, however, with a severe blind spot, because it cannot generally distinguish between $P$ and $Q$, i.e., it does not detect any probability mass in a region where $q(z)$ has small probability."_
>
> 6. About the adversarial ratio regularizer in Section 5: do I understand correctly that it makes no sense for NPE? So the conclusion is the ratio regularizer is useless as opposed to the classical regularizer introduced in section 4? Also, going in the same direction as Reviewer vyzo: why introduce the regularizer if it shows the same (bad) results as the already existing regularizers (thanks for adding the experiments in Figure 12!!). I was also wondering if the regularizer necessary yields a distribution resembling the prior or if it could be something else (question 12).
>
> _For easier comprehensibility, we renamed the section about the adversarial TV regularizer to "Classical Coverage Regularization Can Be Deceptive" to highlight the main finding of this section and we rewrote some its content. We also included the discussion about the ratio regularization in this section. Furthermore, we moved the results of the regularizers from the literature to the main text (Section 6.3) to highlight that bad results from the TV regularizer are not specific to this regularizer but apply generally to classical coverage regularizers._
>
> 7. Notations: The "push-forward" function notation introduced in Remark 3.6 is still very confusing to me. I would really simplify that. In general, as mentioned in my review, I would clarify whether notations are taken from the literature (with references!) or not.
>
> _Note that the concept of the "push-forward" of a probability measure cannot just be simplified. The usual way of representing probability distributions just with probability densities, as customary in machine learning, does not allow providing a "simple" introduction, it rather becomes even more complicated.
> The definition of the push-forward measure can be found in measure theory textbook, e.g. Terence Tao, An introduction to measure theory, American Mathematical Society, 2021 (ISBN 978-1-4704-6640-4). We added a reference to the push-forward definition to the manuscript._

---

> ### Author Response · Authors · 2026-01-19
> **Second Response to Reviewer ojnF**
>
> 8. Clarifications: In my opinion it is still important to define / remind the reader what a discriminative function is in Definition 3.1. A footnote would be enough. Also, the footnote saying that the surrogate model is such that P(x)=Q(X) refers to section 1, but you never explain why we actually have P(X)=Q(X) (or why it is okay to assume that). I suggest adding a sentence in the footnote clarifying that.
>
> _The reason for $P(X)=Q(X)$ is that we sample $Q(X)$ exactly like $P(X)$:
> Firstly, we draw parameters from the prior $\theta \sim P(\Theta)$. Then we plug these parameters into the simulator $x \sim P(X|\Theta=\theta)$. And finally, we use exactly this data in our model $\theta' \sim Q(\Theta|X)$. So $x$ plays both the role as a sample from the forward simulator $P(X)$ and input to the surrogate model $Q(X)$.
> We made the footnote more clear by changing the text to:_
>
> > In SBI, the simulated data $P(X)$ and the input data to the model $Q(X)$ are both generated in the same way: $\theta \sim P(\Theta)$ and then $x \sim P(X|\Theta=\theta)$, hence $P(X)=Q(X)$.
>
> 9. Practical use of the method: You never explicitly mention how many samples are needed to train (and evaluate) the classifier. I second my question 13, it is important to include all experimental details and justify choices. At the very least you should refer to other work, e.g. the l-c2st paper which by the way shows that the sample size for the classifier is not trivial at all and can severely influence the discriminative power of the method. In general, I feel like you could make space to better explain the practical workflow of your method. I do understand that the theoretical framework is part of your contribution, but you are claiming to provide a diagnostic tool that is practical to use, so there should not be any open questions left after reading the paper. As I mentioned in my review, this would also clarify the theory and make the whole thing more convincing (e.g. putting the algorithms in the main paper to illustrate the definitions of coverage and connect both theory and practice).
>
> _The general steps to evaluate the ratio coverage in Section A.2 Setup were reworked:
> First we train a classifier with a binary cross-entropy loss on 1000 data points $(\theta, x)$ from the simulator, which either have the ground truth $\theta \sim p(\theta)$ or an estimated $\theta \sim q(\theta | x)$ from the model. The classifier is a dense network with 5 layer and 64 hidden features and ELU activation. An architecture search was performed and showed similar results in most cases. Since the architecture is relatively lightweight compared to the training of the NPE model and the quality of the resulting ratio coverage plot was satisfactory, no further optimization was necessary._
>
>
> 10. Missing literature and related work: I still think a more explicit connection to local coverage plots (Zhao et al.) is missing. Especially as you mention that L-C2ST "does not specifically address coverage". You could for example just add the citation about L-C2ST you wrote in your response: "is more sensible, precise, and computationally efficient than its concurrent method, local-HPD." As the ratio coverage is more or less equivalent to L-C2ST, that strengthens your claims and makes your proposal more convincing. Also, I do agreee with your response to question 16, it is nice to have a connection to coverage and a interpretable plot in addition to the L-C2ST. However, the last sentence in the "Related work" paragraph suggests that the ratio coverage is stronger ("An estimator with ratio coverage would pass L-C2ST for all x_o"), but it actually goes both ways as both methods are equivalent. Also I do not necessarily agree, it would only pass the L-C2ST for the x_o used for the ratio coverage plot, for any other x_o there is no guarantee. This is where L-C2ST is powerful, as it can test for individual (new) observations x_o. Finally, you mention the related work from Dirmeier et al. at the same place as L-C2ST from Linhart et al. This is confusing: you don't explain what they do, so one might think that they are also part of the L-C2ST work. Please clarify that or remove this reference.
>
> _To avoid confusion in the Related Work section, we made the requested changes:_
> >We addressed the relevant two-sample tests and coverage testing for SBI above; however, we specifically mention the work $\ell$-C2ST Linhart et al. (2023) and for sequential processes SSNL Dirmeier et al. (2024). ℓ-C2ST, which is more precise and computationally efficient than local-HPD Zhao et al. (2021), is similar to ratio coverage because we also estimate a ratio between q(θ | x) and p(θ | x) using the likelihood ratio trick (Hermans et al. (2020); Durkan et al. (2020); Miller et al. (2022); Gutmann & Hyvärinen (2010); Oord et al. (2018); Dalmasso et al. (2020)) or other methods (Miller et al. (2023); Federici et al. (2023); Nguyen et al. (2010); Yao & Domke (2023)).

---

### Decision · Action_Editor_Edz2 · 2026-02-16

**Recommendation:** Accept with minor revision

**Additional Comments:**

The following points do need to be addressed in the final revision:

1. Related work paragraph: The mention of local-HPD comes out of nowhere. Have a brief paragraph on local coverage tests that details this research, which is the back-bone of l-C2ST and the proposed ratio coverage.
2. Clarify the relationship between l-C2ST and the proposed ratio-coverage. Be more precise about what ratio coverage gives in addition to l-c2st and vice versa. When and why one should be used instead of the other in the SBI-workflow? Or maybe they give complementary information?
3. Explain how algorithm 1 connects to definition 3.1.
4. Same for algorithm 2 and definition 3.3
4. Mention the link to the code repository in the main text, e.g. at the beginning of Section 6, or in the contribution paragraph of the introductions.
5. Illustrate definition 3.3 with a graphics. This will make the paper more accessible.
6. Discuss difference to Likelihood-free inference via classification (https://link.springer.com/article/10.1007/s11222-017-9738-6) which also builds on the link between the TV distance and classification.
7. Fig 6: readers may easily think that the the results with the blue (orange) colours of the top and bottom row correspond to each other. To my understanding that would be incorrect. If indeed not correct, I would suggest to use a different colour-scheme.
8. Introduction: "a non-linear function, known as a simulator, that takes in parameters θ and returns a simulated observation x; a" clarify that x is stochastic; a function is normally thought to be a deterministic mapping.

Please respond point by point, stating how and where in the final manuscript the issues above are included/addressed.

**Audience:**

Yes

**Audience Explanation:**

The paper's topic is the evaluation of simulation-based inference (SBI) algorithms, which is of interest to a subset of the TMLR audience.

**Claims And Evidence:**

Yes

**Claims Explanation:**

The recommendation of the reviewers is mixed, but a majority leans to accept. While the final version will need to address remaining issues (see below for a list), the findings of the paper are supported by theory and experiments, and I believe that the paper does make contributions that are worthwhile communicating to the SBI and TMLR audience.

---

> ### Author Response · Authors · 2026-03-03
> **Response to Editor**
>
> We want to thank the Editor and all the reviewers for their contribution to this work and for accepting our manuscript. In the following we address the remaining points:
>
> 1. Related work paragraph: The mention of local-HPD comes out of nowhere. Have a brief paragraph on local coverage tests that details this research, which is the back-bone of l-C2ST and the proposed ratio coverage.
> 2. Clarify the relationship between l-C2ST and the proposed ratio-coverage. Be more precise about what ratio coverage gives in addition to l-c2st and vice versa. When and why one should be used instead of the other in the SBI-workflow? Or maybe they give complementary information?
>
> _The **related work** paragraph (page 3) has been changed to address the mentioned concerns:_
>
> > We addressed the relevant two-sample tests and coverage testing for SBI above; however, we specifically mention the work ℓ-C2ST Linhart et al. (2023) and for sequential processes SSNL Dirmeier et al. (2024). ℓ-C2ST, which is more precise and computationally efficient than its concurrent method local-HPD Zhao et al. (2021), is similar to ratio coverage because we also estimate a ratio between q(θ | x) and p(θ | x) using the likelihood ratio trick (Hermans et al. (2020); Durkan et al. (2020); Miller et al. (2022); Gutmann & Hyvärinen (2010); Oord et al. (2018); Dalmasso et al. (2020)) or other methods (Miller et al. (2023); Federici et al. (2023); Nguyen et al. (2010); Yao & Domke (2023)). ℓ-C2ST and the HPD methodology summarizes the whole information concerning θ into a single scalar or a θ-vector respectively, while this work uses an estimator to improve the statistical power of the coverage plot. So instead of proposing a new method for inspecting local posterior consistency, we show that the widely used coverage plot can be modified to achieve full discriminative power. An estimator with ratio coverage would pass ℓ-C2ST for all $x_o$.
>
> 3. Explain how algorithm 1 connects to definition 3.1.
> 4. Same for algorithm 2 and definition 3.3
>
> _Remarks A.2, A.3 and A.4 were reworked to address the link between the Definitions and Algorithms and were moved from the Appendix to the main text (subsections 3.1 and 3.2 on pages 4 to 6)._
> >**Remark 3.2** (Estimating the coverage plot with samples)
>     Assume that we have an i.i.d. sample $\{z_1,\dots,z_N\}$ from $P$ and an i.i.d. sample $\{z_1',\dots,z_M'\}$ from $Q$.
>     Then we can approximate the coverage plot $\Gamma_g(P,Q)$ as follows.
>     First consider the set of all relevant thresholds:
>     \begin{align}
>         T &:= \{ g(z_n) | n \in [N] \} \cup \{ g(z_m') | m \in [M] \}.
>     \end{align}
>     Then for every $t \in T$ we compute:
>     \begin{align}
>         &\hat p(t) := \\# \{ n \in [N] | g(z_n) \ge t\}/N,
>         &\hat q(t) := \\# \{ m \in [M] | g(z_m') \ge t\}/M,
>     \end{align}
>     and plot the corresponding points for all $t \in T$:
>     \begin{align}
>         \hat \Gamma_g(P,Q) &:= \{ (\hat q(t),\hat p(t)) | t \in T \}.
>     \end{align}
>     See Algorithm 1 and Figure 1 for more information.
>
> >Before estimating the expected conditional coverage plot, note that for every $x \in X$ and $\gamma \in [0,1]$ we always have:
> \begin{align}
>     Q(g(\Theta,x) \ge t_\gamma(x) |X=x) &\ge \gamma, & Q(C_g(\gamma)) \ge \gamma,
> \end{align}
> with equalities under suitable continuity/positivity conditions on the conditionals $Q(\Theta|X=x)$. In those cases we are thus just plotting  $P(C_g(\gamma))$ against $\gamma \in [0,1]$.
> Additionally note that we have the following equivalence for $t\in\bar R$ and $\gamma \in [0,1]$ under suitable continuity/positivity conditions:
>     \begin{align}
>         S(t|x) \le \gamma \qquad\iff\qquad t \ge t_\gamma(x).
>     \end{align}
>     **Remark 3.5** (Estimating the expected conditional coverage plot with samples)
>         For $n \in [N]$ sample $\theta_n \sim P(\Theta)$ and $x_n \sim P(X|\Theta=\theta_n)$ and put $t_n:=g(\theta_n,x_n)$.
>         For $m \in [M]$ further sample $\theta_{n,m} \sim Q(\Theta|X=x_n)$.
>         Then put:
>         \begin{align}
>             \gamma_n:= \frac{1}{M}\sum_{m=1}^M 1[g(\theta_{n,m},x_n) \ge t_n] \approx Q(g(\Theta,x_n) \ge t_n|X=x_n) = S(t_n|x_n).
>         \end{align}
>         Further define for $\gamma \in [0,1]$:
>         \begin{align}
>             \hat F(\gamma) := \frac{1}{N}\sum_{n=1}^N 1[\gamma_n \le \gamma]
>             &\approx
>             \frac{1}{N}\sum_{n=1}^N 1[S(t_n|x_n) \le \gamma] \\
>             &\approx
>             \frac{1}{N}\sum_{n=1}^N 1[t_n \ge t_\gamma(x_n)]  \\
>             &= \frac{1}{N}\sum_{n=1}^N 1[g(\theta_n,x_n) \ge t_\gamma(x_n)] \\
>             &\approx P(g(\Theta,X) \ge t_\gamma(X)).
>         \end{align}
>         With these approximations and the argument in (16) we can thus estimate the expected conditional coverage plot as:
>         \begin{align}
>             \hat\Gamma_g(P\|Q) := \{ ( \gamma,\hat F(\gamma) ) | \gamma \in  [0,1] \} \subset [0,1]\times[0,1].
>         \end{align}
>         See Algorithm 2 and Figure 1 for more information.

---

> ### Author Response · Authors · 2026-03-03
> **Response to Editor**
>
> 5. Mention the link to the code repository in the main text, e.g. at the beginning of Section 6, or in the contribution paragraph of the introductions.
>
> _The link is now included in the beginning of Section 6 (page 10)._
> > Code is available: https://anonymous.4open.science/r/ratio_coverage-E777/.
>
> 6. Illustrate definition 3.3 with a graphics. This will make the paper more accessible.
>
> _Fig. 1 illustrates Definition 3.1 and 3.3 equally, since the figure shows a one-dimensional case which is identical for conditional and unconditional coverage. A direct reference to Figure 1 was added to Remarks 3.2 and 3.5 (see previous responses to points 3 and 4)._
>
> 7. Discuss difference to Likelihood-free inference via classification (https://link.springer.com/article/10.1007/s11222-017-9738-6) which also builds on the link between the TV distance and classification.
>
> _The mentioned Gutmann et al. (2018) paper uses the TV distance in a different way compared to our manuscript. We show that the TV distance is an upper bound on the Coverage Gap while Gutmann et al. (2018) show that a classifier converges to the ratio q(z)/p(z) under $L_1$ norm, hence TV distance, without mentioning Coverage.  e added a reference to the paper in Section 4 (page 8):_
>
> > However, in a practical setting the ratio coverage cannot be calculated directly, because it requires the model distribution q(z) as well as the true posterior distribution p(z). Therefore, we propose to train a classifier to approximate the ratio q(z)/p(z). This approach must be distinguished from Neural Ratio Estimation (NRE), which instead learns the likelihood-to-evidence ratio. Additionally, a single value metrics can also be obtained by calculating the total variation norm between both distributions directly using the learned ratio as similarly proposed in Gutmann et al. (2018).
>
> 8. Fig 6: readers may easily think that the the results with the blue (orange) colours of the top and bottom row correspond to each other. To my understanding that would be incorrect. If indeed not correct, I would suggest to use a different colour-scheme.
>
> _Changed the color scale of Figure 6 on page 13._
>
> 9. Introduction: "a non-linear function, known as a simulator, that takes in parameters θ and returns a simulated observation x; a" clarify that x is stochastic; a function is normally thought to be a deterministic mapping.
>
> _Reformulated the sentence in the Section 1 on page 1._
>
> > Recall that the necessary ingredients for deep learning-based SBI include: a non-linear, stochastic function, known as a simulator, that takes in parameters θ and returns a simulated observation x;